# Compressed Image Generation with Denoising Diffusion Codebook Models

**Guy Ohayon** [* 1]   **Hila Manor** [* 2]   **Tomer Michaeli** [2]   **Michael Elad** [1]

## Abstract

We present a novel generative approach based on Denoising Diffusion Models (DDMs), which produces high-quality image samples *along* with their losslessly compressed bit-stream representations. This is obtained by replacing the standard Gaussian noise sampling in the reverse diffusion with a selection of noise samples from pre-defined codebooks of fixed iid Gaussian vectors. Surprisingly, we find that our method, termed *Denoising Diffusion Codebook Model* (DDCM), retains sample quality and diversity of standard DDMs, even for extremely small codebooks. We leverage DDCM and pick the noises from the codebooks that best match a given image, converting our generative model into a highly effective lossy image codec achieving state-of-the-art perceptual image compression results. More generally, by setting other noise selections rules, we extend our compression method to any conditional image generation task (e.g., image restoration), where the generated images are produced jointly with their condensed bit-stream representations. Our work is accompanied by a mathematical interpretation of the proposed compressed conditional generation schemes, establishing a connection with score-based approximations of posterior samplers for the tasks considered. Code and demo are available on our project's website.

## 1. Introduction

Denoising Diffusion Models (DDMs) (Sohl-Dickstein et al., 2015; Ho et al., 2020) have emerged as an effective tool for generating samples from complex signal distributions (e.g., natural images). Hence, DDMs are commonly leveraged to facilitate a variety of downstream tasks, such as text-to-image synthesis (Ramesh et al., 2021; Rombach et al., 2022; Saharia et al., 2022), editing (Meng et al., 2022; Huberman-Spiegelglas et al., 2024), compression (Theis et al., 2022; Elata et al., 2024; Körber et al., 2024), and restoration (Kawar et al., 2022; Chung et al., 2023). Common to many of these applications is the reliance on iterative sampling from a continuous Gaussian distribution, yielding an unbounded representation space.

This work embarks on the hypothesis that such an infinite representation space is highly redundant. For example, consider any stochastic diffusion generative process with $T = 1000$ sampling steps (e.g., DDPM (Ho et al., 2020)). Suppose that at each timestep, the generative process is restricted to choosing between only two fixed noise realizations. Sampling could then lead to $2^{1000}$ different outputs, an incredibly large number exceeding the estimated amount of atoms in the universe. Thus, in principle, such a process could cover the distribution of natural images densely.

We harness this intuition and propose *Denoising Diffusion Codebook Models* (DDCM), a novel DDM generation scheme for continuous signals, leveraging a discrete and finite representation space. In particular, we first construct a chain of *codebooks*, where each is a sequence of pre-sampled Gaussian noise vectors. These codebooks are constructed once and remain fixed for the entire lifetime of the model. Then, during the generative process, we simply randomly pick the noises from the codebooks instead of drawing them from a Gaussian distribution, as shown in Fig. 2. Since we alter only the sampling process, DDCM can be applied using any pre-trained DDM. Interestingly, we find that our proposed discrete and finite representation space is indeed expressive enough to retain the generative capabilities of standard DDMs, even when using incredibly small codebooks. Since our generative process is entirely governed by the noise *indices* picked during the generation, an important consequence is that every generated image can be perfectly reconstructed by repeating the process with its corresponding indices.

We leverage this property to solve a variety of tasks, using gradient-free noise selection rules to guide the DDCM gener-

*Equal contribution  [1]Faculty of Computer Science, Technion – Israel Institute of Technology, Haifa, Israel  [2]Faculty of Electrical and Computer Engineering, Technion – Israel Institute of Technology, Haifa, Israel. Correspondence to: Guy Ohayon <guyoep@gmail.com>, Hila Manor <hila.manor@campus.technion.ac.il>.

*Proceedings of the 42nd International Conference on Machine Learning*, Vancouver, Canada. PMLR 267, 2025. Copyright 2025 by the author(s).

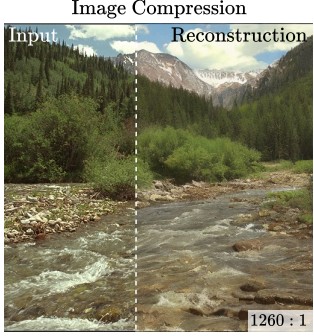 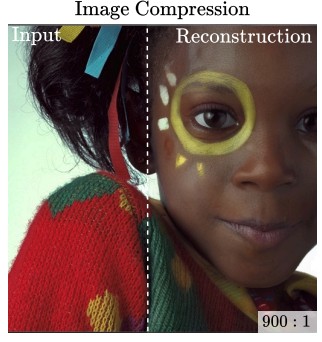 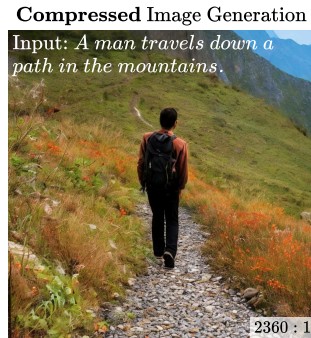 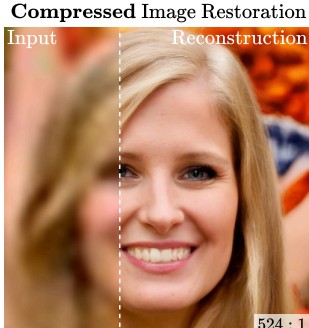

Image Compression · Image Compression · **Compressed** Image Generation · **Compressed** Image Restoration

*Figure 1.* Our proposed scheme (DDCM) produces visually appealing image samples with high compression ratios (bottom-right corners).

ation process. In particular, by choosing the discrete noises to best match a given image, we achieve state-of-the-art perceptual compression results. Moreover, using DDCM with different noise selection rules yields a versatile framework for other *compressed* conditional generation tasks, such as compressed image restoration (see examples in Fig. 1). Finally, we provide a mathematical interpretation of the proposed schemes based on score-based generative modeling with SDEs (Song et al., 2020), showing a connection between our generalized selection rules and approximate posterior sampling for compressed conditional generation.

## 2. Related Work

**Compression.** Image compression has seen significant progress in recent years, with the penetration of neural networks to this domain. Neural methods range from constructing specialized architectures (Ballé et al., 2017; Zhu et al., 2022; Jiang et al., 2023; Jiang & Wang, 2023) to relying on different generative models such as Generative Adversarial Networks (GANs) (Mentzer et al., 2020; Muckley et al., 2023; Iwai et al., 2024) and Variational Autoencoders (VAEs) (Theis et al., 2017). Recent compression methods leverage DDMs and offer high perceptual quality results, by training models from scratch (Yang & Mandt, 2023; Ghouse et al., 2023), fine-tuning existing models (Careil et al., 2023; Körber et al., 2024), or using pre-trained DDMs in a zero-shot manner (without further training) (Theis et al., 2022; Elata et al., 2024). Current solutions in the latter category are highly computationally demanding, either due to their communication schemes (e.g., reverse channel coding (Theis & Ahmed, 2022; Theis et al., 2022)) or their need to perform thousands of denoising operations (Elata et al., 2024). Our work falls into this last category, offering a novel and highly effective compression scheme with a fast bit-stream communication method and computational demands that match standard use of DDMs.

**Discrete Generative Modeling for Continuous Data.** Recent works have explored discrete generative modeling of continuous data distributions. These employ various discrete representations, such as vector quantized latent tokens (Wu et al., 2024) or hierarchical modeling schemes that gradually refine each generated sample (Yang, 2025). DDCM offers a new such discrete generative framework, building on the exceptional achievements of DDMs.

**Conditional Image Generation with Pre-Trained DDMs.** Pre-trained DDMs are commonly utilized for solving conditional image generation tasks, such as image restoration (Kawar et al., 2022; Lugmayr et al., 2022; Wang et al., 2023b; Chung et al., 2023; Lin et al., 2024; Song et al., 2023; Cohen et al., 2024b; Yue & Loy, 2024; Raphaeli et al., 2025; Man et al., 2025) and editing (Meng et al., 2022; Huberman-Spiegelglas et al., 2024; Cohen et al., 2024a; Manor & Michaeli, 2024). In this work we address these tasks from the lens of DDCM, where the conditional samples are generated along with their compressed bit-streams.

**Compressed Image Generation.** The task of compressed image generation (generating images directly in their compressed form) has been previously explored. Kang et al. (2019) trained an unconditional GAN (Goodfellow et al., 2014) to synthesize JPEG representations. Rajesh et al. (2023) trained a text-conditional GAN in the JPEG domain. In this work, we use DDMs instead of GANs and introduce a novel compressed image representation space different than that of JPEG. Our approach is compatible with any pre-trained DDM without requiring additional training.

**Compressed Conditional Image Generation.** Our work also addresses the problem of compressed *conditional* generation (see Sec. 6), in which the compressed output is generated in accordance with a given input condition, such as a text prompt, a noisy image, or other forms of guidance. In this context, Liu et al. (2021) proposed a method based on optimal transport, subject to an informational bot-

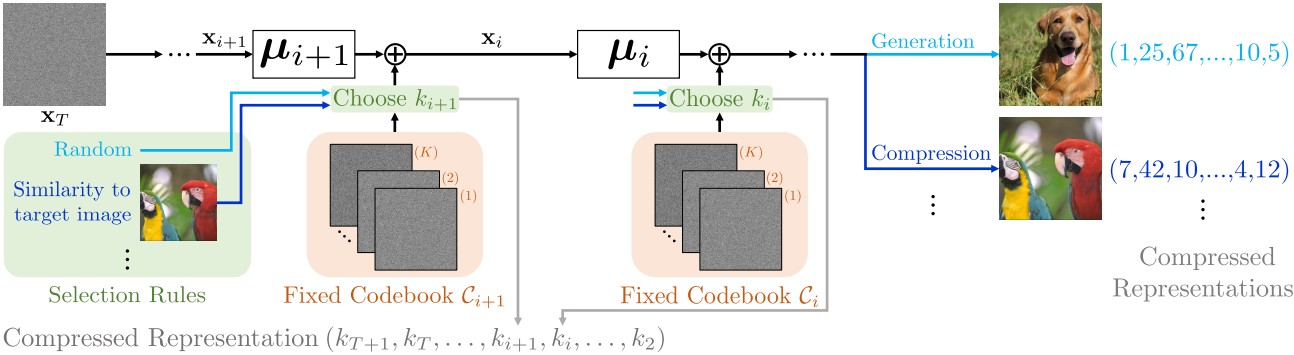

*Figure 2.* **Method overview.** DDCM replaces the standard Gaussian noises in DDPM sampling with a selection of noise samples from pre-defined codebooks of fixed iid Gaussian vectors. This retains the high-quality generative properties of standard DDMs, while producing the results along with their compressed representations. By choosing the discrete noises according to different selection rules, DDCM can perform a variety of conditional image generation tasks. Our highly condensed bit-stream representation is especially effective for image compression, leading to state-of-the-art results.

tleneck constraint. Specifically, their approach involves computing an optimal transport map from the distribution of input conditions to the distribution of target signals, while constraining the entropy of an intermediate latent random variable (i.e., the code). This method requires both the inputs and outputs to reside within the same metric space. In contrast, our approach naturally accommodates more complex types of input-output pairs, such as those involving different modalities (e.g., text and image).

## 3. Background

Diffusion models (Sohl-Dickstein et al., 2015; Ho et al., 2020; Song et al., 2020) generate samples from a data distribution $p_0$ by *reversing* a diffusion process that gradually adds random noise to samples from the data. Specifically, the diffusion process starts with $\mathbf{x}_0 \sim p_0$ and produces the chain $\mathbf{x}_0, \mathbf{x}_1, \ldots, \mathbf{x}_T$ via

$$\mathbf{x}_i = \sqrt{\alpha_i}\mathbf{x}_{i-1} + \sqrt{1 - \alpha_i}\mathbf{z}_i, \quad \mathbf{z}_i \sim \mathcal{N}(\mathbf{0}, \boldsymbol{I}), \quad (1)$$

where $\alpha_1, \ldots, \alpha_T > 0$ are some time-dependent constants. The above is a time-discretization of a Variance Preserving (VP) SDE (Song et al., 2020). Then, samples from the data distribution $p_0$ are generated by solving the corresponding reverse-time VP SDE (Anderson, 1982; Song et al., 2020), i.e., by gradually *denoising* samples, starting from $\mathbf{x}_T \sim \mathcal{N}(\mathbf{0}, \boldsymbol{I})$. In this paper we adopt Denoising Diffusion Probabilistic Models (DDPMs) (Ho et al., 2020), which propose the generative process ($i = T, \ldots, 1$)

$$\mathbf{x}_{i-1} = \boldsymbol{\mu}_i(\mathbf{x}_i) + \sigma_i \mathbf{z}_i, \text{ where} \quad (2)$$

$$\boldsymbol{\mu}_i(\mathbf{x}_i) = \frac{1}{\sqrt{\alpha_i}}(\mathbf{x}_i + (1 - \alpha_i)\boldsymbol{s}_i(\mathbf{x}_i)), \quad (3)$$

$\mathbf{z}_i \sim \mathcal{N}(\mathbf{0}, \boldsymbol{I})$, $\sigma_i = \sqrt{1 - \alpha_i}$, and $\boldsymbol{s}_i(\mathbf{x}_i)$ denotes the *score* $\nabla_{\mathbf{x}_i} \log p_i(\mathbf{x}_i)$ of the probability density function $p_i(\mathbf{x}_i)$.

Such a score function $\boldsymbol{s}_i(\mathbf{x}_i)$ is typically learned via *denoising score matching* (Vincent, 2011; Song & Ermon, 2019; Song et al., 2020; Ho et al., 2020), where a model $\hat{\mathbf{x}}_{0|i}$ is trained to predict $\mathbf{x}_0$ from $\mathbf{x}_i$ (i.e., a denoiser), and using the well-known equation (Robbins, 1956; Miyasawa et al., 1961; Stein, 1981)

$$\boldsymbol{s}_i(\mathbf{x}_i) = \frac{\sqrt{\bar{\alpha}_i}\hat{\mathbf{x}}_{0|i} - \mathbf{x}_i}{1 - \bar{\alpha}_i}, \quad (4)$$

where $\bar{\alpha}_i := \prod_{s=1}^{i} \alpha_s$. This generative process is applicable to both pixel space (Dhariwal & Nichol, 2021) and latent space (Rombach et al., 2022) diffusion models, by employing a VAE-based encoder-decoder.

## 4. Denoising Diffusion Codebook Models

**Method.** Equation (2) depicts the standard DDPM sampling approach, where the added noise is sampled from a continuous Gaussian distribution. DDCM instead uses a discrete noise space, by limiting each sampling step to choose from $K$ constant noise realizations, fixed separately for each step. Formally, for each $i = 2, \ldots, T+1$ we define a codebook of $K$ entries

$$\mathcal{C}_i = \left[\boldsymbol{z}_i^{(1)}, \boldsymbol{z}_i^{(2)}, \ldots, \boldsymbol{z}_i^{(K)}\right], \quad (5)$$

where each $\mathcal{C}_i(k) := \boldsymbol{z}_i^{(k)}$ is sampled independently from $\mathcal{N}(\mathbf{0}, \boldsymbol{I})$ and remains fixed throughout the model's lifetime. Then, we modify the DDPM sampling process (2), replacing the noise $\mathbf{z}_i$ by a randomly selected codebook entry,

$$\mathbf{x}_{i-1} = \boldsymbol{\mu}_i(\mathbf{x}_i) + \sigma_i \mathcal{C}_i(k_i), \quad (6)$$

where the process is initialized with $\mathbf{x}_T = \mathcal{C}_{T+1}(k_{T+1})$, $k_i \sim \text{Unif}(\{1, \ldots, K\})$, and sampling step $i = 1$ does not

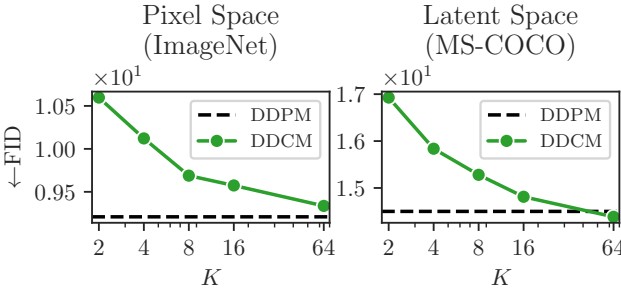

*Figure 3.* **Comparing DDPM with DDCM for different codebook sizes** $K$. Interestingly, DDCM with $K = 64$ achieves similar FID to DDPM, suggesting that the continuous representation space of DDPM (DDCM with $K = \infty$) is highly redundant. We use a class-conditional ImageNet model ($256 \times 256$) for pixel space, and the text-conditional SD 2.1 model ($768 \times 768$) for latent space, with prompts from MS-COCO. The $K$ axis is in log-scale.

involve noise addition. This random selection procedure is depicted in the generation branch in Fig. 2. Importantly, running the generative process (6) with a given sequence of noise vectors $\{\mathcal{C}_i(k_i)\}_{i=2}^{T+1}$ always produces the same output image. Thus, as depicted in the bottom part of Fig. 2, the sequence of indices $k_{T+1}, \ldots, k_2$ can be considered a losslessly compressed bit-stream representation of each generated image.

**Experiments.** While DDPM is equivalent to DDCM with $K = \infty$, the first question we address is whether DDCM maintains the synthesis capabilities of DDPM for relatively small $K$ values. We compare the performance of DDPM with that of DDCM using $K \in \{2, 4, 8, 16, 64\}$ for sampling from pre-trained pixel and latent space models. We compute the Fréchet Inception Distance (FID) (Heusel et al., 2017) to evaluate the generation performance. In App. A we report additional metrics and provide qualitative comparisons. For pixel space generation, we use a class-conditional DDM trained on ImageNet $256 \times 256$ (Deng et al., 2009; Dhariwal & Nichol, 2021), and apply classifier guidance (CG) (Dhariwal & Nichol, 2021) with unit scale. We use the 50k validation set of ImageNet as the reference dataset, and sample 10k class labels randomly to generate the images. For latent space, we use Stable Diffusion (SD) 2.1 (Rombach et al., 2022) trained on $768 \times 768$ images and apply classifier-free guidance (CFG) with scale 3 (equivalent to $w = 2$ in (Ho & Salimans, 2021)). As the reference dataset, we randomly select 10k images from MS-COCO (Lin et al., 2014; Chen et al., 2015) along with one caption per image, and use those captions as prompts for sampling.

As shown in Fig. 3, DDCM achieves similar FID scores to DDPM at $K = 64$, suggesting that the Gaussian representation space of DDPM is redundant. In the next sections we leverage our new representation space to solve a variety of

tasks, including image compression and compressed image restoration.

## 5. Image Compression with DDCM

**Method.** Since sampling with DDCM yields compact bit-stream representations, a natural endeavor is to harness DDCM for compressing real images. In particular, to compress an image $\mathbf{x}_0$, we leverage the predicted $\hat{\mathbf{x}}_{0|i}$ (Eq. (4)) at each timestep $i$ and compute the residual error from the target image, $\mathbf{x}_0 - \hat{\mathbf{x}}_{0|i}$. Then, we guide the sampling process towards $\mathbf{x}_0$ by selecting the codebook entry that maximizes the inner product with this residual,

$$k_i = \arg\max_{k \in \{1, \ldots, K\}} \langle \mathcal{C}_i(k), \mathbf{x}_0 - \hat{\mathbf{x}}_{0|i} \rangle, \qquad (7)$$

where the size of the first codebook $\mathcal{C}_{T+1}$ is $K = 1$. This process is depicted as the compression branch in Fig. 2, where the resulting set of chosen indices $\{k_i\}_{i=2}^{T+1}$ is the compressed bit-stream representation of the given image. Section 6 sheds more light on this choice of the noise selection from the perspective of score-based generative models (Song et al., 2020). As in Sec. 4, decompression follows standard DDCM sampling (6), re-selecting the stored indices instead of picking them randomly. When using latent space DDMs (e.g., SD), we first encode $\mathbf{x}_0$ into the latent domain, perform all subsequent operations in this domain, and decode the result with the decoder.

The bit rate of this approach is determined by the size of the codebooks $K$, and the number of sampling timesteps $T$. Specifically, the bit-stream length is given by $(T - 1)\log_2(K)$. Therefore, the bit rate can be reduced by simply decreasing the number of codebooks, or by using a smaller number of timesteps at generation, e.g., by skipping every other step, or by using specific timestep intervals (see App. B.4). In the approach described so far, the length of the bitstream increases logarithmically with $K$, making it computationally demanding to increase the bit rate. For instance, even for $K = 8192$, $T = 1000$ and $768 \times 768$ images our BPP is approximately 0.022. Thus, to produce higher bit rates, we propose to *refine* the noise selected at timestep $i$ by employing matching pursuit (MP) (Mallat & Zhang, 1993). Specifically, at each step $i$, we construct the chosen noise as a convex combination of $M$ elements from $\mathcal{C}_i$, gathered in a greedy fashion to best correlate with the guiding residual $\mathbf{x}_0 - \hat{\mathbf{x}}_{0|i}$ (as in Eq. (7)). The resulting convex combination involves $M - 1$ quantized scalar coefficients, chosen from a finite set of $C$ values taken from $[0, 1]$. Therefore, the resulting length of the bit-stream is given by $(T - 1)(\log_2(K)M + C(M - 1))$, such that $M = 1$ is similar to our standard compression scheme, and the length of the bit-stream increases linearly with $M$ and $C$. We apply this algorithm when the absolute bits number crosses $(T - 1) \cdot \log_2(2^{13})$. Further details are available in App. B.5.

| Original | BPG | PSC-P | PerCo | **Ours** ($K = 256$) |
|---|---|---|---|---|

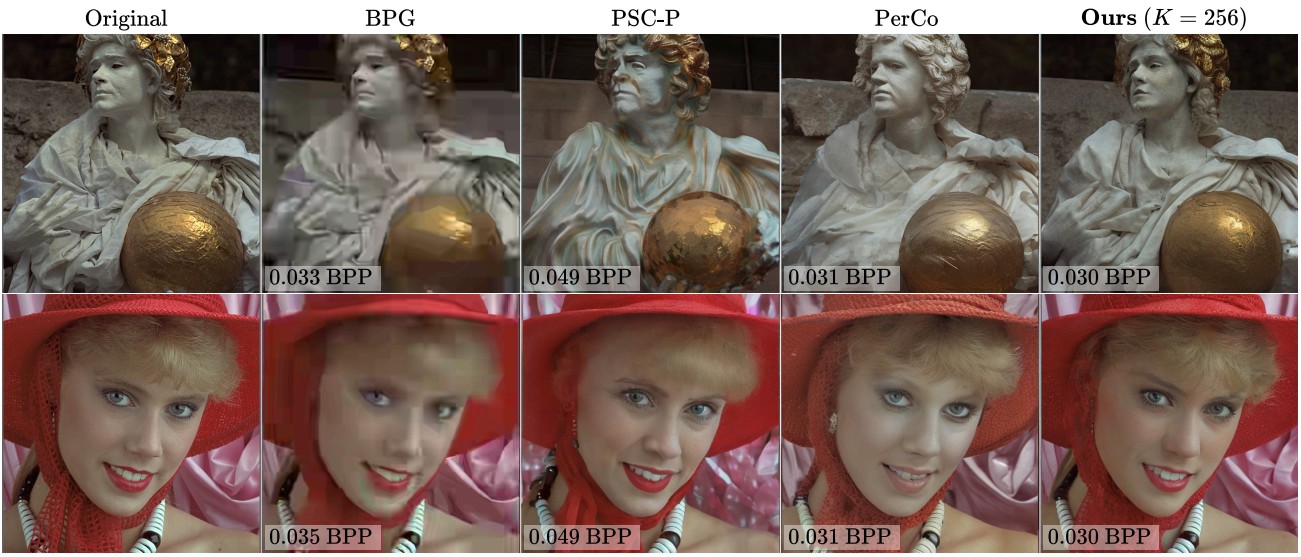

*Figure 4.* **Qualitative image compression results.** The presented images are taken from the Kodak24 ($512 \times 512$) dataset. Our codec produces highly realistic outputs, while maintaining better fidelity to the original images compared to previous methods.

**Experiments.** We evaluate our compression method on Kodak24 (Franzen, 1999), DIV2K (Agustsson & Timofte, 2017), ImageNet 1K $256 \times 256$ (Deng et al., 2009; Pan et al., 2020), and CLIC2020 (Toderici et al., 2020). For all datasets but ImageNet, we center crop and resize all images to $512 \times 512$. We compare to numerous competing methods, both non-neural and neural, and both zero-shot, fine-tuning based, and training based approaches. For the ImageNet dataset, we use the unconditional pixel space ImageNet $256 \times 256$ model of Dhariwal & Nichol (2021), and compare our results to BPG (Bellard, 2018), HiFiC (Mentzer et al., 2020), IPIC (Xu et al., 2024), and two PSC (Elata et al., 2024) configurations, distortion-oriented (PSC-D) and perception-oriented (PSC-P). For all other datasets, we use SD 2.1 $512 \times 512$ (Rombach et al., 2022) and compare to BPG, PSC-D, PSC-P, ILLM (Muckley et al., 2023), PerCo (SD) (Körber et al., 2024; Careil et al., 2023), and twoCRDR (Iwai et al., 2024) configurations, distortion-oriented (CRDR-R) and perception-oriented (CRDR-R). PSC shares the same pre-trained model as ours, while PerCo (SD) requires additional fine-tuning. For our method, we apply SD 2.1 unconditionally, as we saw no improvement by adding prompts (see further details in App. B.6). We assess our method for several options of $T$, $K$, $M$, and $C$ to control the bit rate. See further details in App. B. We evaluate distortion with PSNR and LPIPS (Zhang et al., 2018) and perceptual quality with FID (Bińkowski et al., 2018). For ImageNet, FID is computed against the entire 50k $256 \times 256$ validation set. For the smaller datasets we follow Mentzer et al. (2020) and compute the FID over extracted image patches. Specifically, for DIV2K and CLIC2020 we extract

$128 \times 128$ sized patches, and for Kodak we use $64 \times 64$.

As shown on the rate-distortion and rate-perception planes in Fig. 5, our compression scheme dominates previous methods on the rate-perception-distortion tradeoff (Blau & Michaeli, 2019) for lower bit rates, surpassing both the perceptual quality (FID) and distortion (PSNR and LPIPS) of previous methods. For instance, our FID scores are lower than those of all other methods at around 0.1 BPP, while, for the same BPP, our distortion performance is better than the perceptually-oriented methods (e.g., PerCo, PSC-P, and IPIC). However, our method under-performs at the highest bit rates, especially when using SD. When using a latent DDM such as SD, we aim to compress the latent encoding of a given image, rather than the image itself. Since encoding and decoding an image typically leads to a distorted reconstruction, the distortion of the compression method is bounded by that of the encoder-decoder. This leads to a performance ceiling for any latent-space-based compression method (Körber et al., 2024; Elata et al., 2024). We therefore include in Fig. 5 the "SD 2.1 Encoder-Decoder bound", which corresponds to the distortion resulting from encoding and decoding the original images using the VAE of SD 2.1, without any additional compression. The qualitative comparisons in Fig. 4 further demonstrate our superior perceptual quality, where, even for extreme bit rates, our method maintains similarity to the original images in fine details. See App. B for more details and results.

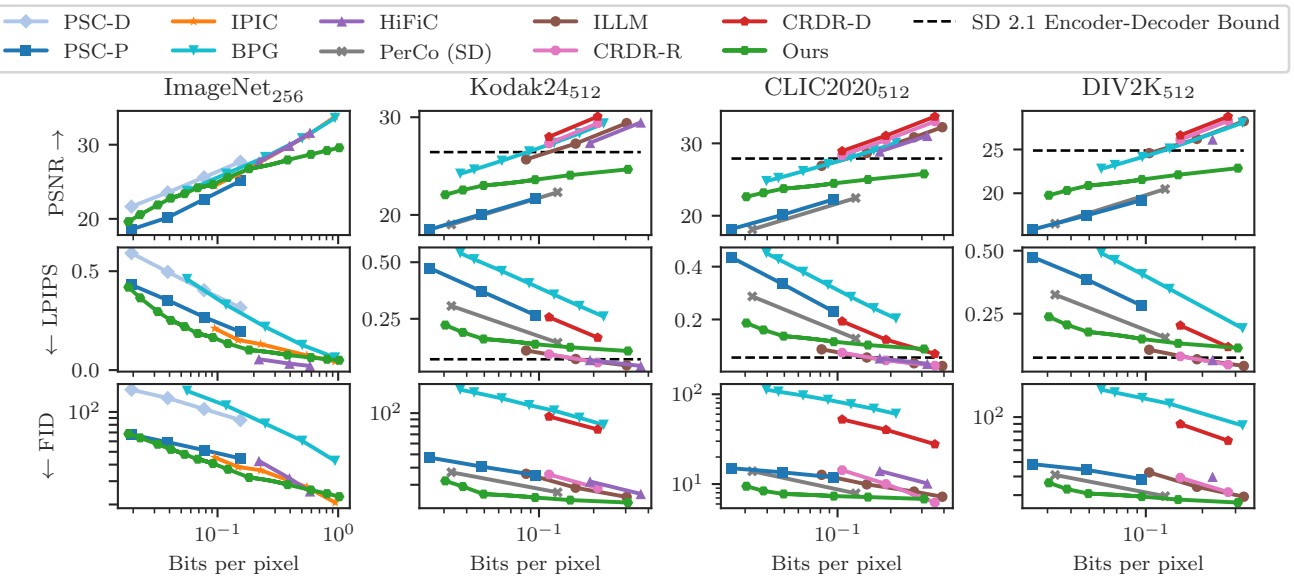

*Figure 5.* **Compression quantitative evaluation.** We compare the perceptual quality (FID) and distortion (PSNR, LPIPS) achieved at different BPPs. The image sizes of each dataset is denoted next to its name. Our method produces the best perceptual quality at most BPPs. Importantly, this is while we also attain lower distortion compared to perceptually-oriented methods (e.g., PSC-P and PerCo (SD)). For the three rightmost datasets, note that our approach, PSC-P, and PerCo (SD) use the latent space Stable Diffusion 2.1 model, so its encoder-decoder imposes a distortion bound. Thus, we report the distortion attained by simply passing the images through this encoder-decoder (dashed line).

## 6. Compressed Conditional Generation

We showed that DDCM can be used as an image codec by following a simple index selection rule, guiding the generated image towards a target one. Here, we generalize this scheme to any *conditional* generation task, considering the more broad framework of *compressing* conditionally generated samples. This is a particularly valuable framework in scenarios where the input condition $\mathbf{y}$ is bit rate intensive, e.g., where $\mathbf{y}$ is a degraded image and the goal is to produce a *compressed* high-quality reconstruction of it. To the best of our knowledge, this task, which we name *compressed* conditional generation, has only been thoroughly explored for text input conditions (Rajesh et al., 2023).

A naive solution to this task could be to simply compress the outputs of any existing conditional generation scheme. Here we propose a novel end-to-end solution that generates the outputs *directly* in their compressed form. Importantly, note that our approach in Sec. 4 requires the condition $\mathbf{y}$ for decompressing the bit-stream. While this is not a stringent requirement when the condition is lightweight (e.g., a text prompt), this approach is less suitable when storage of the condition signal itself requires a long bit-stream. The solutions we propose in this section enable decoding the bit-stream without access to $\mathbf{y}$.

**Compressed Conditional Generation with DDCM.** We propose generating a conditional sample by choosing the indices $k_i$ in Eq. (6) via

$$k_i = \underset{k\in\{1,...,K\}}{\arg\min}\ \mathcal{L}(\mathbf{y}, \mathbf{x}_i, \mathcal{C}_i, k), \qquad (8)$$

instead of picking them randomly. Here, $\mathcal{L}(\mathbf{y}, \mathbf{x}_i, \mathcal{C}_i, k)$ can be any loss function that attains a lower value when $\mathcal{C}_i(k)$ directs the generative process towards an image that matches $\mathbf{y}$. For example, for the loss

$$\mathcal{L}_{\mathrm{P}}(\mathbf{y}, \mathbf{x}_i, \mathcal{C}_i, k) = \|\mathcal{C}_i(k) - \sigma_i \nabla_{\mathbf{x}_i} \log p_i(\mathbf{y}|\mathbf{x}_i)\|^2 \quad (9)$$

we obtain the following result (see proof in App. C):

**Proposition 6.1.** *Suppose that image samples are generated via process* (6)*, and the indices $k_i$ are chosen according to Eq.* (8) *with $\mathcal{L} = \mathcal{L}_{\mathrm{P}}$. Then, when $K \to \infty$, such a generative process becomes a discretization of a probability flow ODE over the posterior distribution $p_0(\mathbf{x}_0|\mathbf{y})$.*

In other words, Proposition 6.1 implies that for the loss $\mathcal{L}_{\mathrm{P}}$, increasing $K$ leads to more accurate sampling from the posterior $p_0(\mathbf{x}_0|\mathbf{y})$, though this results in longer bit-streams. Thus, as long as we have access to $\nabla_{\mathbf{x}_i} \log p_i(\mathbf{y}|\mathbf{x}_i)$ (or an approximation of it) $\mathcal{L}_{\mathrm{P}}$ may serve as a sensible option for solving a compressed conditional generation task with DDCM. Interestingly, we show in App. C.2 that our compression scheme from Sec. 5 is a special case of the proposed

compressed conditional generation method, with $\mathbf{y} = \mathbf{x}_0$ and $\mathcal{L} = \mathcal{L}_{\mathrm{P}}$.

### 6.1. Compressed Posterior Sampling for Image Restoration

Our compressed conditional sampling approach can be utilized for solving inverse problems via posterior sampling. Specifically, we consider inverse problems of the form $\mathbf{y} = \boldsymbol{A}\mathbf{x}_0$, where $\boldsymbol{A}$ is some linear operator. We restrict our attention to *unconditional* diffusion models and solve the problems in a "zero-shot" manner (similarly to Kawar et al. (2022); Chung et al. (2023); Wang et al. (2023b)). To generate conditional samples, we propose using the loss

$$\mathcal{L}(\mathbf{y}, \mathbf{x}_i, \mathcal{C}_i, k) = \|\mathbf{y} - \boldsymbol{A}(\boldsymbol{\mu}_i(\mathbf{x}_i) + \sigma_i \mathcal{C}_i(k))\|^2. \quad (10)$$

Note that Eq. (10) attains a lower value when $\sigma_i \boldsymbol{A}\mathcal{C}_i(k)$ points in the direction that perturbs $\boldsymbol{A}\boldsymbol{\mu}_i(\mathbf{x}_i)$ towards $\mathbf{y}$. Thus, our conditional generative process aims to produce a reconstruction $\hat{\mathbf{x}}$ that satisfies $\boldsymbol{A}\hat{\mathbf{x}} \approx \mathbf{y}$, implying that we approximate posterior sampling (Ohayon et al., 2023). Notably, when assuming that $p_i(\mathbf{y}|\mathbf{x}_i)$ is a multivariate normal distribution centered around $\boldsymbol{A}\mathbf{x}_i$ (as in (Jalal et al., 2021)), the chosen codebook noise $\mathcal{C}_i(k)$ approximates the gradient $\nabla_{\mathbf{x}_i} \log p_i(\mathbf{y}|\mathbf{x}_i)$ and Eq. (10) becomes a proxy of Eq. (9).

Following (Chung et al., 2023; Wang et al., 2023b), we implement our method using the unconditional ImageNet $256 \times 256$ DDM trained by Dhariwal & Nichol (2021). We fix $K = 4096$ for all codebooks, resulting in a compressed bit-stream of approximately $0.183$ BPP for each generated image. We compare our method with DPS (Chung et al., 2023) and DDNM (Wang et al., 2023b) on two noiseless tasks: colorization and $4\times$ super-resolution (using the bicubic kernel). We evaluate these methods using their official implementations and the same DDM. We additionally compress the outputs of DPS and DDNM to assess whether such a naive approach would yield better results. To do so, we adopt our proposed compression scheme (from Sec. 5), employing the same unconditional ImageNet DDM and using $K = 4096$ noises per codebook.

Qualitative and quantitative results are reported in Fig. 6. As expected, due to the rate-perception-distortion tradeoff (Blau & Michaeli, 2019), we observe that compressing the outputs of DPS and DDNM harms either their perceptual quality (FID), or their distortion (PSNR), or both. This is while our method achieves superior perceptual quality compared to both DPS and DDNM, including their compressed versions. While our method achieves slightly worse PSNR, this is expected due to the perception-distortion tradeoff (Blau & Michaeli, 2018). See App. C.3 for more details.

### 6.2. Compressed Real-World Face Image Restoration

Real-world face image restoration is the practical task of restoring any degraded face image, without any knowledge of the corruption process it has gone through (Wang et al., 2021; Gu et al., 2022; Wang et al., 2022; Zhou et al., 2022; Wang et al., 2023c; Lin et al., 2024; Yue & Loy, 2024; Chen et al., 2024b; Ohayon et al., 2025). We propose a novel method capable of optimizing any no-reference image quality assessment (NR-IQA) measure at test time (e.g., NIQE (Mittal et al., 2013)), without relying on gradients.

Specifically, at each timestep $i$, we start by picking two indices – one that promotes high perceptual quality, $k_{i,P}$, and another that promotes low distortion, $k_{i,D}$. Then, we choose between $k_{i,P}$ and $k_{i,D}$ the index that better optimizes a desired balance of the perception-distortion tradeoff (Blau & Michaeli, 2018). Formally, letting $\boldsymbol{r}(\mathbf{y}) \approx \mathbb{E}[\mathbf{x}_0|\mathbf{y}]$ denote the approximate Minimum Mean-Squared-Error (MMSE) estimator of this task, we pick $k_{i,D}$ via

$$k_{i,D} = \underset{k \in \{1,...,K\}}{\arg\max} \langle \mathcal{C}_i(k), \boldsymbol{r}(\mathbf{y}) - \hat{\mathbf{x}}_{0|i} \rangle. \quad (11)$$

Note that this index selection rule is similar to that of our standard compression, replacing $\mathbf{x}_0$ in Eq. (7) with $\boldsymbol{r}(\mathbf{y})$. This choice of indices in DDCM would lead to a reconstructed estimate of the MMSE solution $\boldsymbol{y}(\mathbf{y})$, yielding blurry results with low distortion (Blau & Michaeli, 2018). In contrast, *randomly* picking a sequence of indices in DDCM would produce a high quality sample from the data distribution $p_0$. Therefore, we randomly choose $k_{i,P} \sim \mathrm{Unif}(\{1, \ldots, K\})$. Then, we use the DDM and compute $\hat{\mathbf{x}}_{0|i-1}$ for each index $k \in \{k_{i,D}, k_{i,P}\}$ separately, denoting each result accordingly by $\hat{\mathbf{x}}_{0|i-1}^{(k)}$. The final index is picked to optimize the perception-distortion tradeoff via

$$k_i = \underset{k \in \{k_{i,D}, k_{i,P}\}}{\arg\min} \mathrm{MSE}\left(\boldsymbol{r}(\mathbf{y}), \hat{\mathbf{x}}_{0|i-1}^{(k)}\right) + \lambda Q\left(\hat{\mathbf{x}}_{0|i-1}^{(k)}\right), \quad (12)$$

where $Q(\cdot)$ can be *any* NR-IQA measure, even a nondifferentiable one. In App. C.4 we explain our choice to set $\boldsymbol{r}(\mathbf{y})$ as an MMSE estimator.

We assess our approach choosing $\boldsymbol{r}(\mathbf{y})$ as the FFHQ (Karras et al., 2019) $512 \times 512$ approximate MMSE model trained by Yue & Loy (2024). We set $\lambda = 1$ and optimize three different $Q(\cdot)$ measures: NIQE, CLIP-IQA$^+$ (Wang et al., 2023a), and TOPIQ (Chen et al., 2024a) adapted for face images by PyIQA (Chen & Mo, 2022). We utilize the FFHQ $512 \times 512$ DDM of Yue & Loy (2024) with $T = 1000$ sampling steps and $K = 4096$ for all codebooks. We compare our approach against the state-of-the-art methods PMRF (Ohayon et al., 2025), DifFace (Yue & Loy, 2024), and BFRffusion (Chen et al., 2024b), using the standard evaluation datasets CelebA-Test (Karras et al., 2018; Wang et al., 2021), LFW-Test (Huang et al., 2008), WebPhoto-Test (Wang et al., 2021), and WIDER-Test (Zhou et al.,

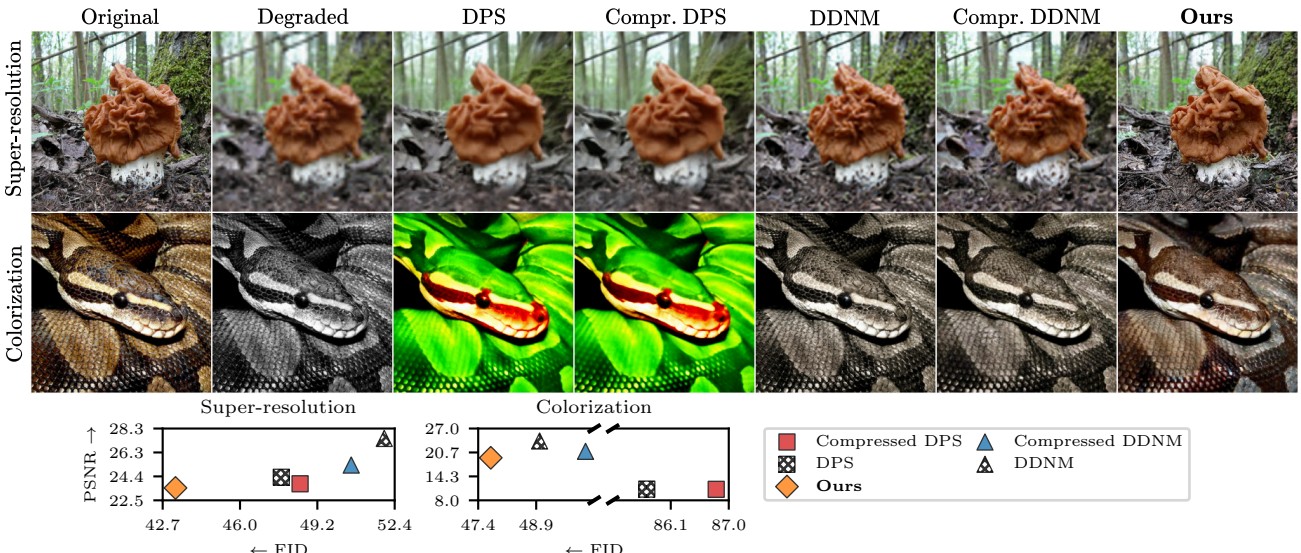

*Figure 6.* **Comparison of zero-shot posterior sampling image restoration methods.** Our approach achieves better perceptual quality compared to previous methods, while maintaining competitive PSNR and automatically producing compressed bit-stream representations for each restored image.

2022). We use PSNR to measure the distortion of the outputs produced for the CelebA-Test dataset, and measure the ProxPSNR (Ohayon et al., 2025; Man et al., 2025) for the other datasets, which lack the clean original images. Perceptual quality is measured by NIQE, CLIP-IQA$^+$, TOPIQ-FACE, and additionally FD$_{\text{DINOv2}}$ (Stein et al., 2023) to assess our generalization performance to a common quality measure which we do not directly optimize. Finally, as in Sec. 6.1, we *compress* each evaluated method using our standard compression scheme, adopting the same FFHQ DDM with $K = 4096$ and $T = 1000$.

The results for the WIDER-Test dataset are reported in Fig. 7 (see App. C.4 for the other datasets). Our approach clearly optimizes each quality measure effectively and generalizes well according to the FD$_{\text{DINOv2}}$ scores. This is also confirmed visually, where all of our solutions produce high-quality images with less artifacts compared to previous methods. While our approach shows slightly worse distortion, this is once again expected due to the perception-distortion tradeoff (Blau & Michaeli, 2018).

## 7. Discussion

We introduced DDCM, a novel generative approach for DDMs that produces high-quality image samples jointly with their lossless compressed bit-stream representations. We found that DDCM achieves comparable generative performance to DDPM, even when the codebooks are extremely small. We leveraged DDCM to solve several compressed image generation tasks, including image compression and compressed restoration, where we achieved state-of-the-art

results. Besides image restoration, our compressed conditional generation framework can be used for any type of diffusion guidance, e.g., for text-conditional generation. We demonstrate this option in Apps. C.5 and C.6, introducing new classifier-based and classier-free guidance methods that do not use **y** for decompression. Moreover, we present in App. C.7 preliminary results for compressed image editing using DDCM, by decompressing an image using a desired edit text prompt.

While our empirical results are encouraging, our work does not explain theoretically why DDCM sampling and our simple index selection strategies work so effectively. We encourage future works to investigate the principles behind the success of our methods. Moreover, when operating in latent space, our codec's performance is bounded by the underlying VAE, particularly at higher bit rates. Our results at higher bit rates could also be improved through better approaches than our current matching pursuit inspired solution. Additionally, our compression efficiency could be improved through entropy coding of the selected indices, potentially reducing bit rates without sacrificing quality. Lastly, all DDCM-based solutions can be improved further by optimizing the codebooks, e.g., through dictionary learning.

While DDCM produces compressed image representations in the form of indices, it is worth noting that the codebook entries can be interpreted as discrete image tokens, which are chosen autoregressively. This implies that DDCM can be interpreted as an image tokenization method, forming a "language" for images. Such a perspective opens the door to intriguing possibilities, such as using the tokenized

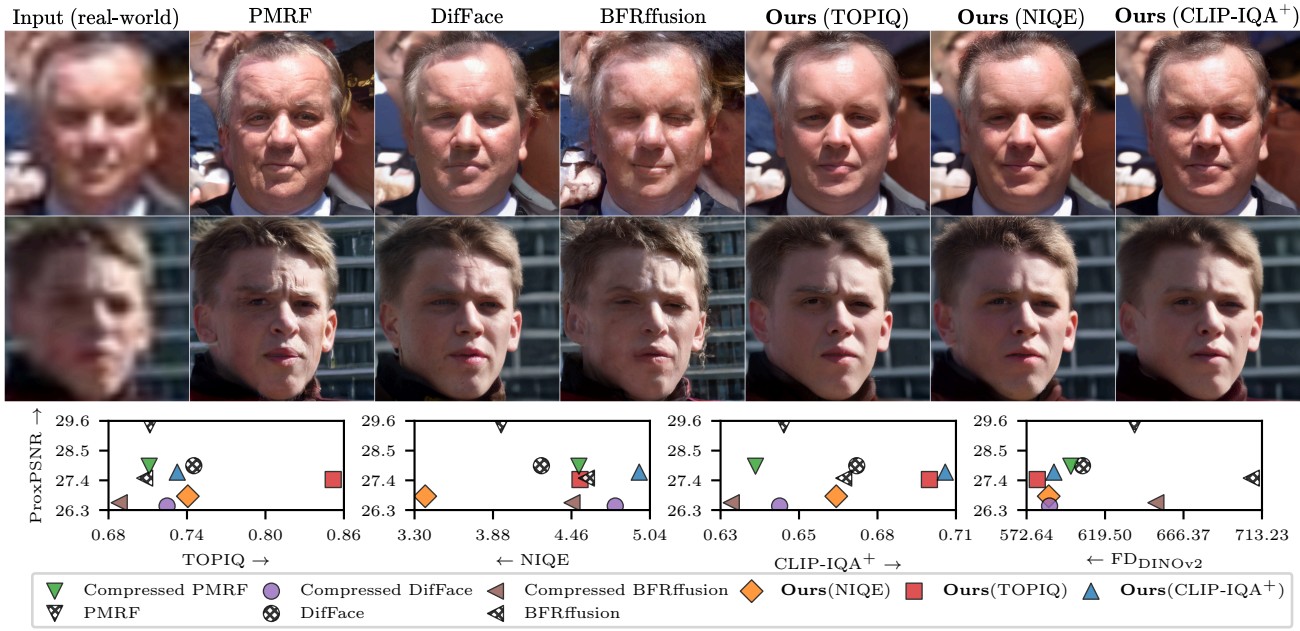

*Figure 7.* **Comparing real-world face image restoration methods on the WIDER-Test dataset**. We successfully optimize the NR-IQA measures and produce appealing output perceptual quality with less artifacts compared to previous methods.

image representation as a natural text-conditioning mechanism for unconditional diffusion models. For instance, one could train a transformer to predict the sequence of codebook indices corresponding to an image given its text description. Such a transformer can then be used to generate text-conditional images, by predicting the corresponding sequence of DDCM codebook indices.

To conclude, our work demonstrates promising results across numerous tasks while providing opportunities for both theoretical analysis and practical improvements.

## Acknowledgements

This research was partially supported by the Israel Science Foundation (ISF) under Grants 2318/22, 951/24 and 409/24, and by the Council for Higher Education – Planning and Budgeting Committee. We thank Noam Elata and Matan Kleiner for assisting with our figures and experiments.

## Impact Statement

This paper presents work whose goal is to advance the field of Machine Learning. There are many potential societal consequences of our work, none which we feel must be specifically highlighted here.

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

# A. Additional DDCM Evaluation

Figure 8 provides additional quantitative comparisons between DDPM and DDCM, using different $K$ values. Specifically, we compute the Kernel Inception Distance (KID) (Bińkowski et al., 2018), as well as the Fréchet Distance and Kernel Distance evaluated in the feature space of DINOv2 (Stein et al., 2023; Oquab et al., 2024). These quantitative results remain consistent with the ones presented in Sec. 4, showing that DDCM with small $K$ values is comparable with DDPM. Figures 9 and 10 show numerous outputs from both DDPM and DDCM with small values of $K$, demonstrating the sample quality and diversity produced by the latter for such $K$ values. We use Torch Fidelity (Obukhov et al., 2020) to compute the perceptual quality measures.

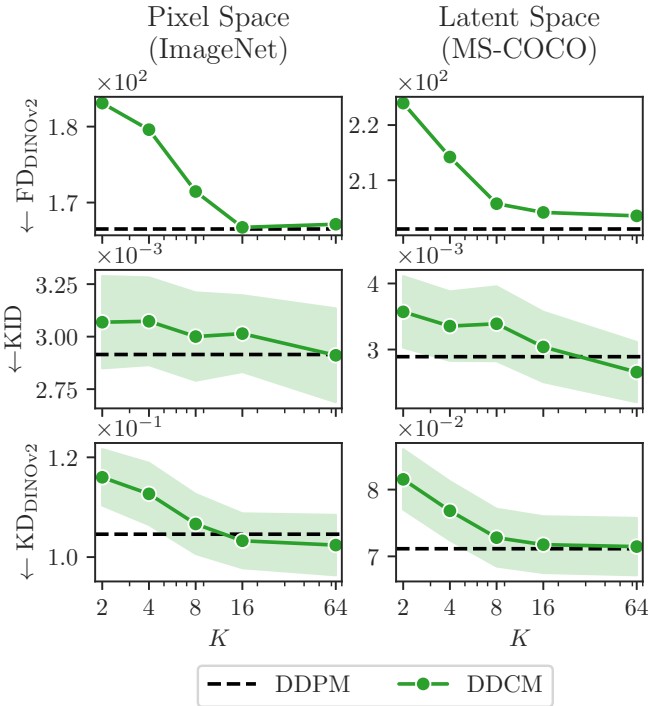

*Figure 8.* **Comparing DDPM with DDCM for different codebook sizes $K$.** As in Sec. 4, DDPM and DDCM with $K = 64$ (sometimes even $K = 16$) achieve similar generative performance, suggesting that the continuous representation space of DDPM (DDCM with $K = \infty$) is highly redundant. We use a class-conditional ImageNet model ($256 \times 256$) for pixel space, and the text-conditional SD 2.1 model ($768 \times 768$) for latent space. The $K$ axis is in log-scale.

A cat is sitting under an umbrella near a yellow box.

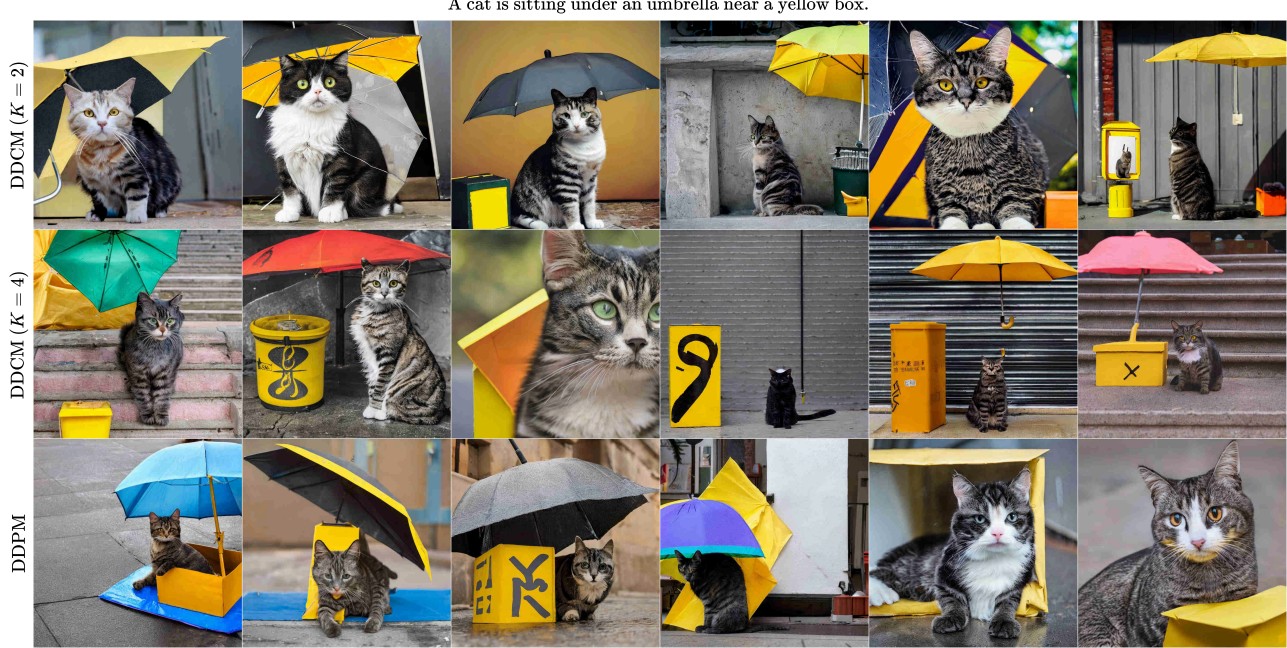

A smiling male tennis player swinging a racket

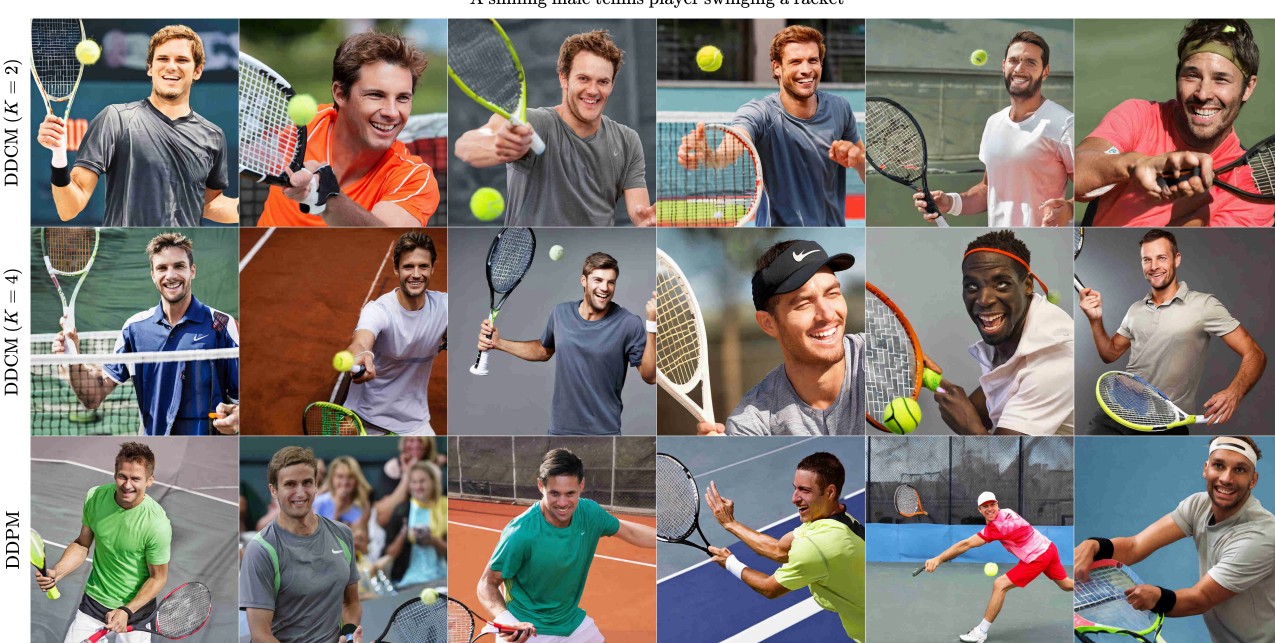

*Figure 9.* **Qualitative comparison of sample quality and diversity between DDCM and DDPM.** We generate multiple samples for each prompt, using the $768 \times 768$ SD 2.1 model.

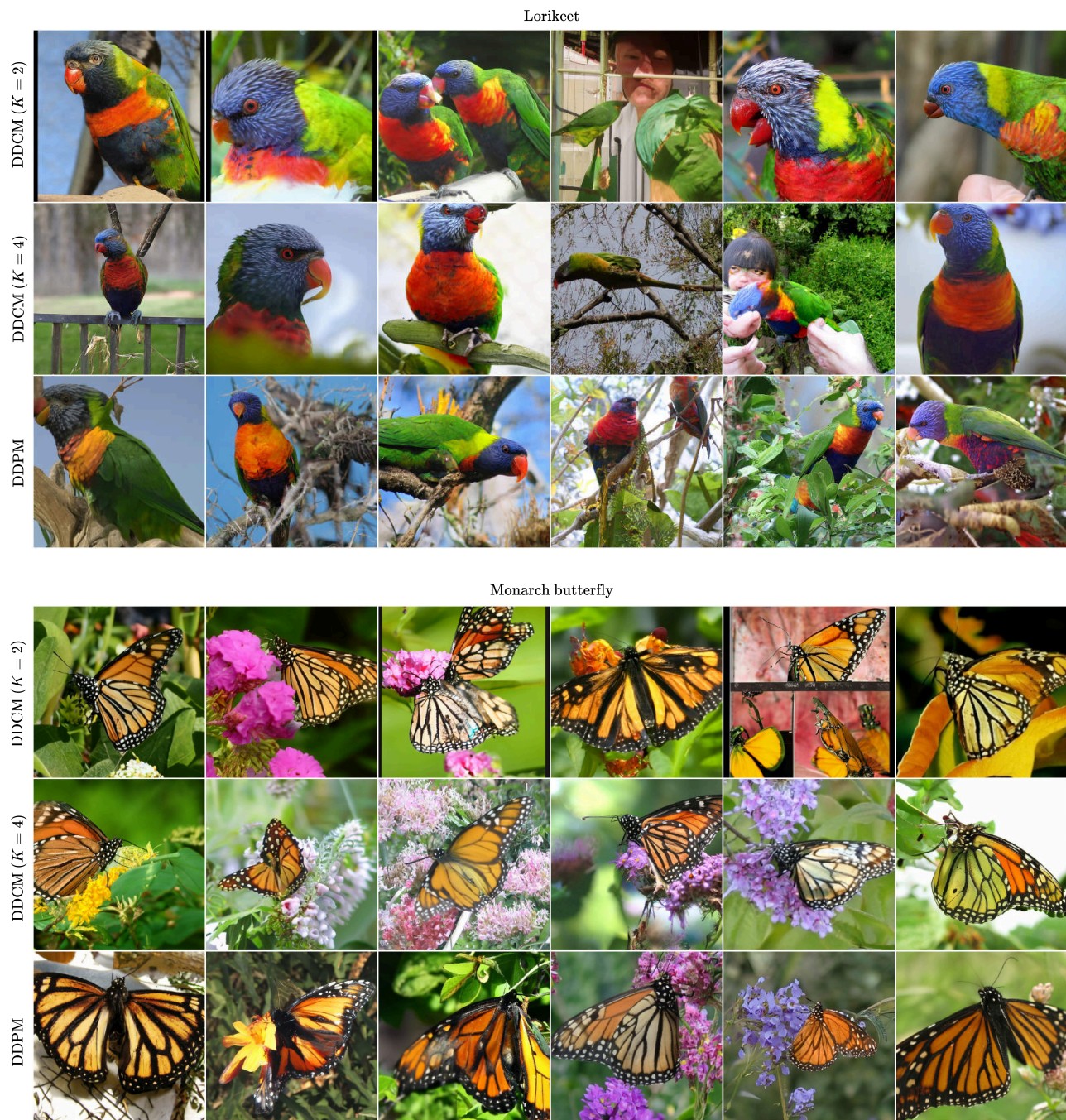

*Figure 10.* **Qualitative comparison of sample quality and diversity between DDCM and DDPM.** We generate multiple samples for each class, using the conditional $256 \times 256$ ImageNet model.

# B. Image Compression Supplementary

We compute all distortion and perceptual quality measures using Torch Metrics (which relies on Torch Fidelity (Obukhov et al., 2020)).

## B.1. Experiment Configurations

We specify here the different configurations used for the compression experiments in Sec. 5.

- In our $256 \times 256$ experiments we use the `256x256_diffusion_uncond.pt` checkpoint from the official GitHub repository. In our $768 \times 768$ and $512 \times 512$ experiments we use Stable Diffusion 2.1 with the `stabilityai/stable-diffusion-2-1` and `stabilityai/stable-diffusion-2-1-base` official checkpoints from Hugging Face, respectively. The different $K$, $M$, $C$ and $T$ values for each of the $512 \times 512$ and the $768 \times 768$ experiments plotted in Figs. 5 and 15 are summarized in Tab. 1.

*Table 1.* Image compression experiments configurations.

| Model | Image Resolution | $T$ | $K$ | $M$ | $C$ |
|---|---|---|---|---|---|
| Pixel Space DDM | $256 \times 256$ | 1000 | 64, 128, 256, 4096 | 1 | - |
| | | 1000 | 2048 | 2, 3, 4, 5 | 3 |
| | | 500 | 128, 512 | 1 | - |
| | | 300 | 16, 32, 128, 512 | 1 | - |
| Latent Space DDM | $512 \times 512$ | 1000 | 256, 1024, 8192 | 1 | - |
| | | 1000 | 2048 | 2, 3, 6 | 3 |
| | $768 \times 768$ | 1000 | 16, 32, 64, 256, 1024, 8192 | 1 | - |
| | | 1000 | 2048 | 2, 3, 6 | 3 |
| | | 500 | 16, 32, 64, 256, 1024, 8192 | 1 | - |
| | | Adapted 500 (App. B.4) | 16, 32, 64, 256, 1024, 8192 | 1 | - |

- For PSC-D and PSC-R we use the same pre-trained ImageNet $256 \times 256$ model as ours in the $256 \times 256$ experiments, and the same Stable Diffusion 2.1 model in the $512 \times 512$ experiments. We adopt the default hyper-parameters of the method as described by the authors (Elata et al., 2024), setting the number of measurements to $12 \cdot 2^i$ for $i = 0, \dots, 8$.

- For IPIC we adopt the official implementation using the ELIC codec with five bit rates, combined with DPS sampling for decoding with $T = 1000$ steps and $\zeta \in \{0.3, 0.6, 0.6, 1.2, 1.6\}$, as recommended by the authors.

- For BPG we considered quality factors $q \in \{51, 50, 48, 46, 42, 40, 38, 36, 34, 32, 30\}$.

- For HiFiC we test the low, medium and high quality regimes, using the checkpoints available in the official GitHub repository.

- PerCo (SD) is tested using the three publicly available Stable Diffusion 2.1 fine-tuned checkpoints from their Official GitHub repository, using the default hyper-parameters.

- For ILLM we use the MS-ILLM pre-trained models available in the official GitHub implementation. For the $512 \times 512$ image size experiments we use `msillm_quality_X`, X = 2, 3, 4. For the $768 \times 768$ image size experiments we use `msillm_quality_X`, X = 2, 3 and `msillm_quality_vloY`, Y = 1, 2.

- CRDR-D and CRDR-R are evaluated using quality factors of $\{0, 1, 2, 3, 4\}$, where CRDR-D uses $\beta = 0$ and CRDR-R uses $\beta = 3.84$, as recommended in the paper.

## B.2. Additional Evaluations

In Figs. 11, 12, 13 and 14 we provide additional qualitative comparisons on the Kodak24 ($512 \times 512$) and ImageNet ($256 \times 256$) datasets. We additionally compare our method on images of size $768 \times 768$, and present the results in Fig. 15. Our method is implemented as before, while using a Stable Diffusion 2.1 model trained on the appropriate image size (see App. B.1).

## B.3. Numerical Results

Tabs. 2, 3, 4 and 5 include the numerical results that appear in Fig. 5. Additionally, Tabs. 6 and 7 include the numerical results for Fig. 15.

## B.4. Decreasing the Bit Rate via Timestep Sub-Sampling

As mentioned in Sec. 5, decreasing the bit rate of our compression scheme can be accomplished in two ways. The first option is to reduce $K$, which sets the number of bits required to represent each communicated codebook index. The second option relates to the number of generation timesteps, which sets the total number of communicated indices. Specifically, DDMs trained for $T = 1000$ steps can still be used to generate samples with fewer steps, by skipping alternating timesteps and modifying the variance in Eq. (2). Thus, we leverage such timestep sub-sampling in DDCM to shorten the compressed bit-stream. We find that pixel space DDMs yields good results with this approach, while the latent space models struggle to produce satisfying perceptual quality.

Thus, for latent space models we propose a slightly different timestep sub-sampling scheme. Specifically, we keep $T = 1000$ sampling steps at inference and set different $K$ values for different subsets of timesteps. We choose $K = 1$ for a subset of $L$ sampling steps, and $K > 1$ for the rest $T - L$ steps. Thus, our compression scheme only optimizes $T - L$ steps and necessitates transmitting only $T - L$ indices. The rest $L$ indices correspond to codebooks that contain only one vector, and thus do not affect the bit rate.

We use $T = 1000$, set the same codebook size $K > 1$ for every timestep $i \in \{899\ldots, 400\}$, and use $K = 1$ for all other steps. We compare our proposed method against the aforementioned naive timestep skipping approach with $T = 500$ sampling steps and the same $K > 1$, which attains the same bit rate as our proposed alternative.

Quantitative results are shown in Fig. 15, where our timestep adapted method is denoted by *Ours Adapted*. Our adapted approach achieves better perceptual quality compared to the naive one, at the expense of a slightly hindered PSNR.

## B.5. Increasing the Bit Rate via Matching Pursuit

In Sec. 5 we briefly explain how to achieve higher bit rates with our method, by refining each selected noise via a matching pursuit inspired solution. Formally, at each timestep $i$ we iteratively refine the selected noise by linearly combining it with $M - 1$ other noises from the codebook. We start by picking the first noise index $k_i$ according to Eq. (7). Then, we set $k_i^{(1)} = k_i$, $\gamma_i^{(1)} = 1$, and $\tilde{z}_i^{(1)} = \mathcal{C}_i(k_i)$, and pick the next indices and coefficients $(m = 1, \ldots, M - 1)$ via

$$k_i^{(m+1)}, \gamma_i^{(m+1)} = \underset{k \in \{1,\ldots,K\},\, \gamma \in \Gamma}{\arg\max} \left\langle \gamma \tilde{z}_i^{(m)} + (1 - \gamma)\, \mathcal{C}_i(k), \mathbf{x}_0 - \hat{\mathbf{x}}_{0|i} \right\rangle. \tag{13}$$

The noise vector $\tilde{z}_i^{(m+1)}$ is then updated via

$$\tilde{z}_i^{(m+1)} \leftarrow \gamma_i^{(m+1)} \tilde{z}_i^{(m)} + \left(1 - \gamma_i^{(m+1)}\right) \mathcal{C}_i\left(k_i^{(m+1)}\right), \tag{14}$$

$$\tilde{z}_i^{(m+1)} \leftarrow \frac{\tilde{z}_i^{(m+1)}}{\text{std}\left(\tilde{z}_i^{(m+1)}\right)}, \tag{15}$$

where $\text{std}(\mathbf{z})$ is the empirical standard deviation of the vector $\mathbf{z}$. We use the resulting vector $\tilde{z}_i^{(M)}$ as the noise in Eq. (6) to produce the next $\mathbf{x}_{i-1}$, and repeat the above process iteratively. Note that setting $M = 1$ is equivalent to our standard compression scheme.

In our experiments, the set of coefficients $\Gamma$ is a subset of $(0, 1]$, containing $C = |\Gamma|$ values that are evenly spaced in this range. We pick $C = 3$ and assess $M \in \{2, 3, 6\}$ for the latent space model experiments, and $M \in \{2, 3, 4, 5\}$ for the pixel space model experiments.

## B.6. Assessing the Effectiveness of Text Prompts in Compression using Text-to-Image Latent Space DDMs

Stable Diffusion 2.1 is a text-to-image generative model, which both PerCo (SD) and PSC leverage for their compression approach. Specifically, both of these methods start by generating a textual caption for every target image using BLIP-2 (Li

et al., 2023), and feed the captions as prompts to the SD model. In our case, we find that using such prompts hinders the compression quality. Specifically, we follow the same automatic captioning procedure as in PerCo (SD) and PSC, using the `Salesforce/blip2-opt-2.7b-coco` checkpoint of BLIP-2 from Hugging Face. We then continue with our standard compression approach, where the denoiser is used with standard classifier-free guidance (CFG). Note that using text prompts requires transmitting additional bits that serve as a compressed version of the text. Specifically, we use BLIP-2 with a maximum of $L = 32$ word tokens, each picked from a dictionary containing a total of 30,524 words. Thus, at most $32 \cdot \log_2(30524) \approx 480$ bits are added to the bit-stream in our method.

We assess this text-conditional approach on the $512 \times 512$ SD 2.1 DDM, using CFG scales of $3, 6$. We compare the performance of this conditional approach with that of the unconditional one we used in Sec. 5. The results in Fig. 16 show a disadvantage for using text-prompts for compression with our method.

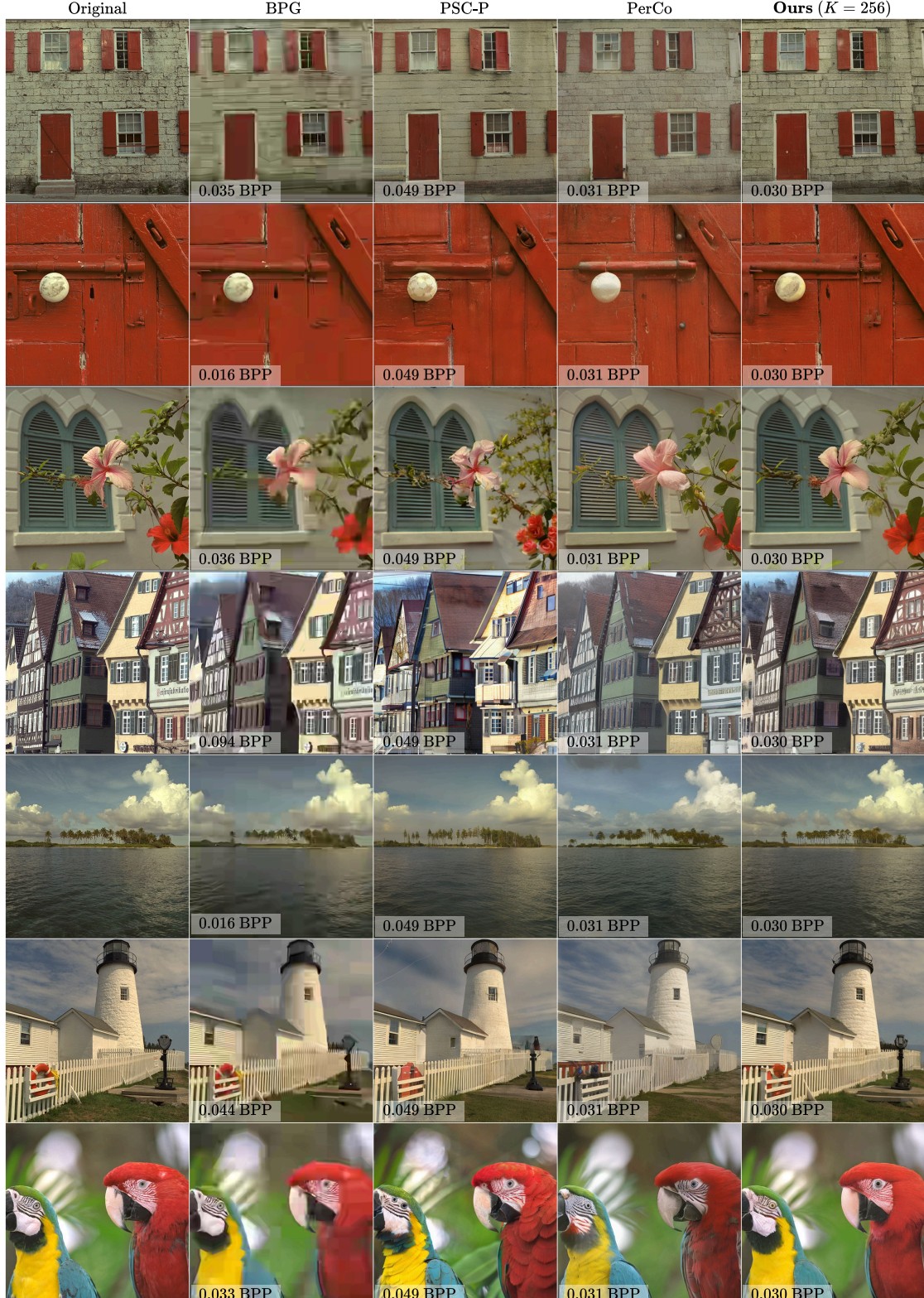

*Figure 11.* **Qualitative extreme image compression results.** The presented images are taken from the Kodak24 dataset, cropped to $512 \times 512$ pixels. Our compression scheme produces highly realistic decompressed outputs, while maintaining better fidelity to the original images compared to previous methods.

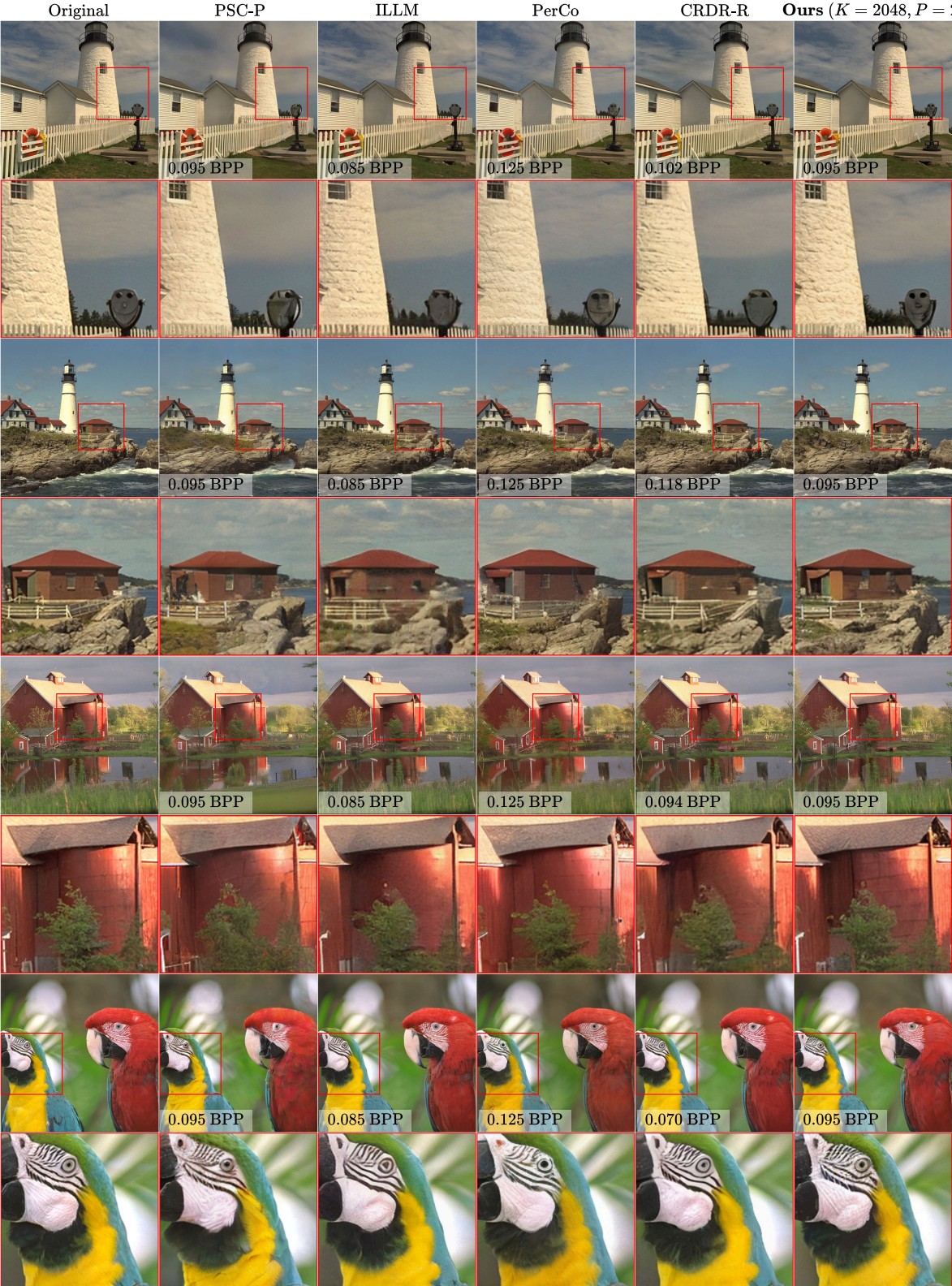

*Figure 12.* **Qualitative image compression results.** The presented images are taken from the Kodak24 dataset, cropped to $512 \times 512$ pixels. Our compression scheme produces highly realistic decompressed outputs, while maintaining better fidelity to the original images compared to previous methods.

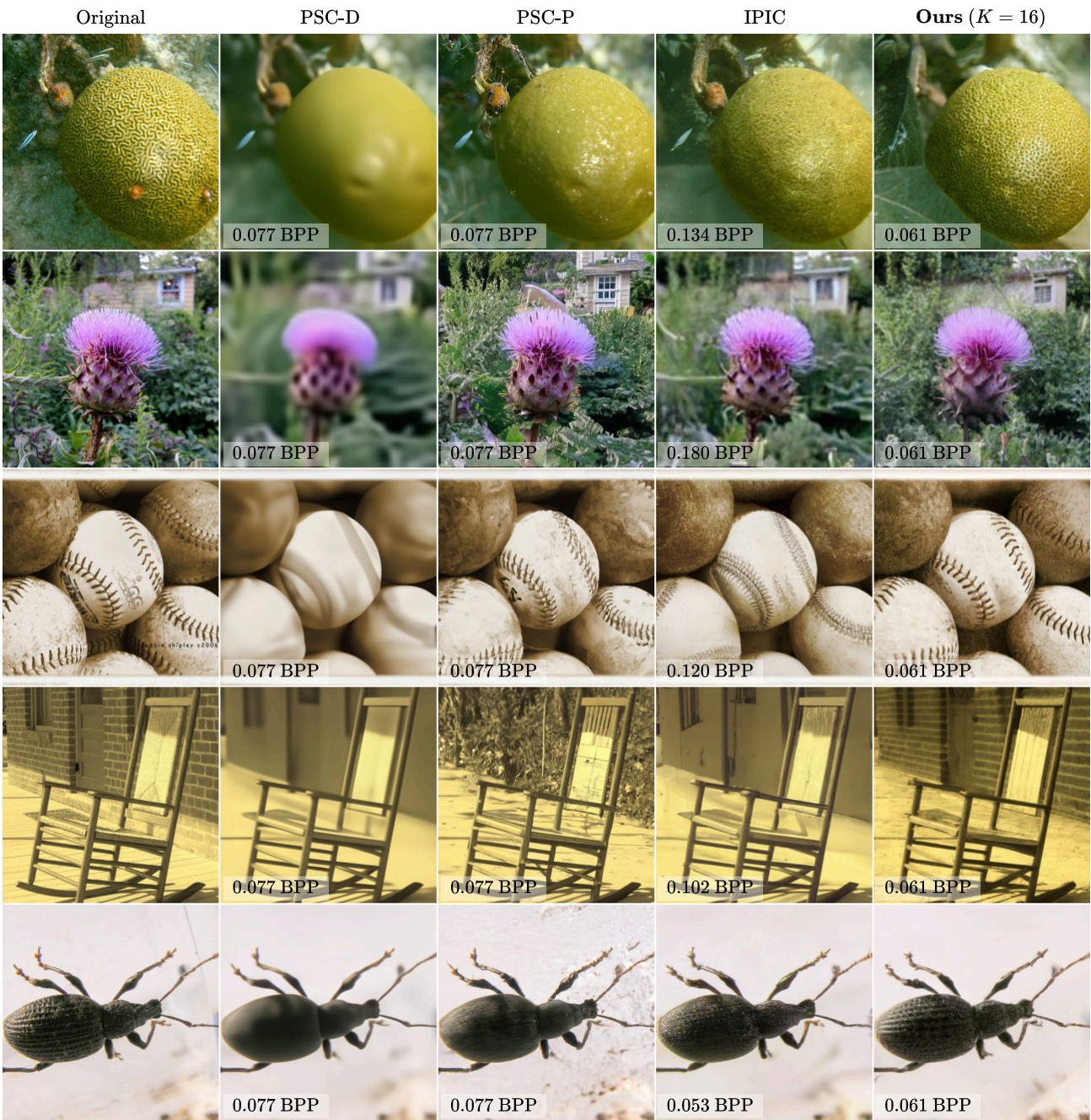

*Figure 13.* **Qualitative image compression results.** the presented images are taken from the ImageNet $256 \times 256$ dataset. Compared to previous methods, our compression scheme produces higher perceptual quality and better fidelity to the original images.

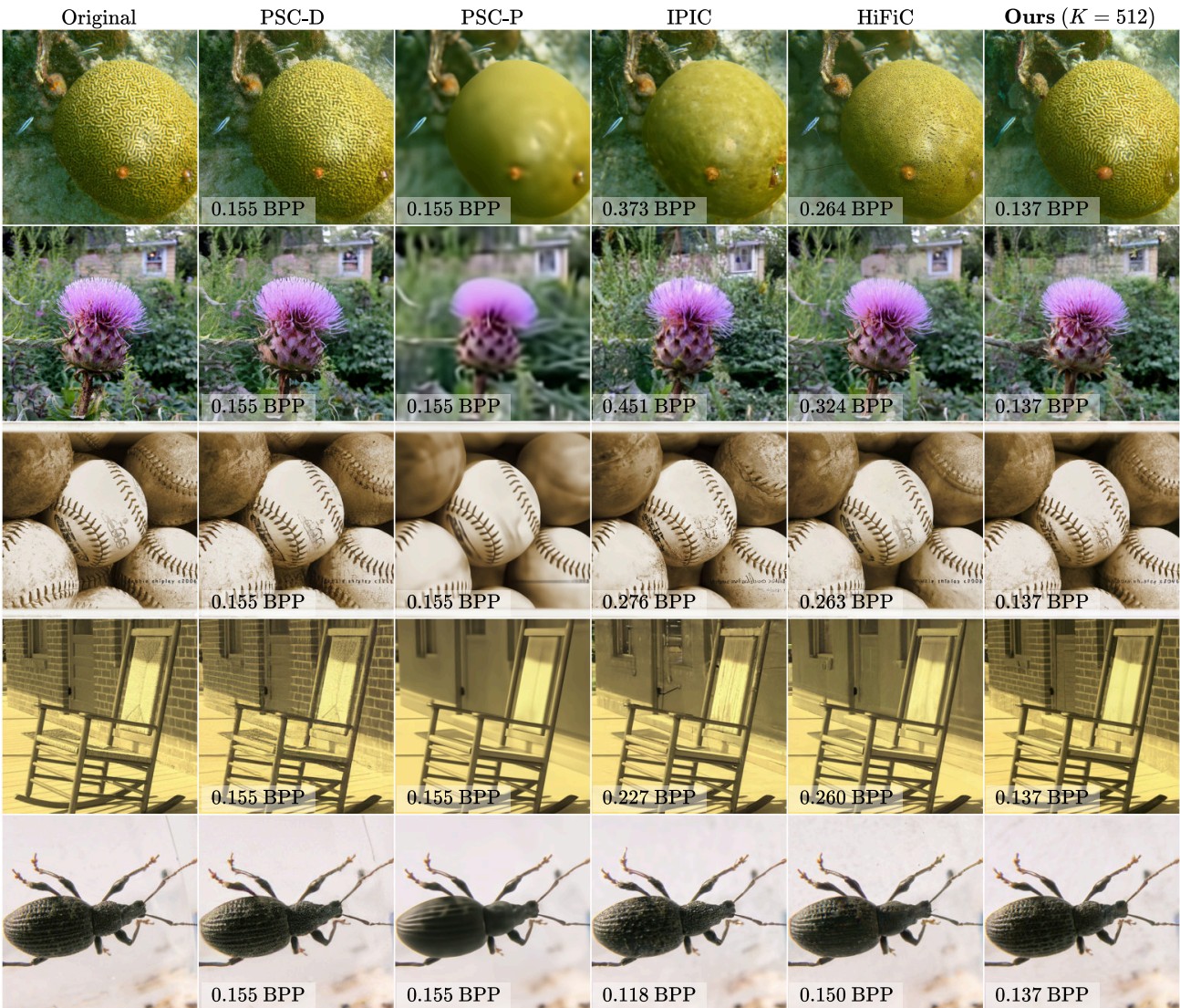

*Figure 14.* **Qualitative image compression results.** the presented images are taken from the ImageNet $256 \times 256$ dataset. Compared to previous methods, our compression scheme produces higher perceptual quality and better fidelity to the original images.

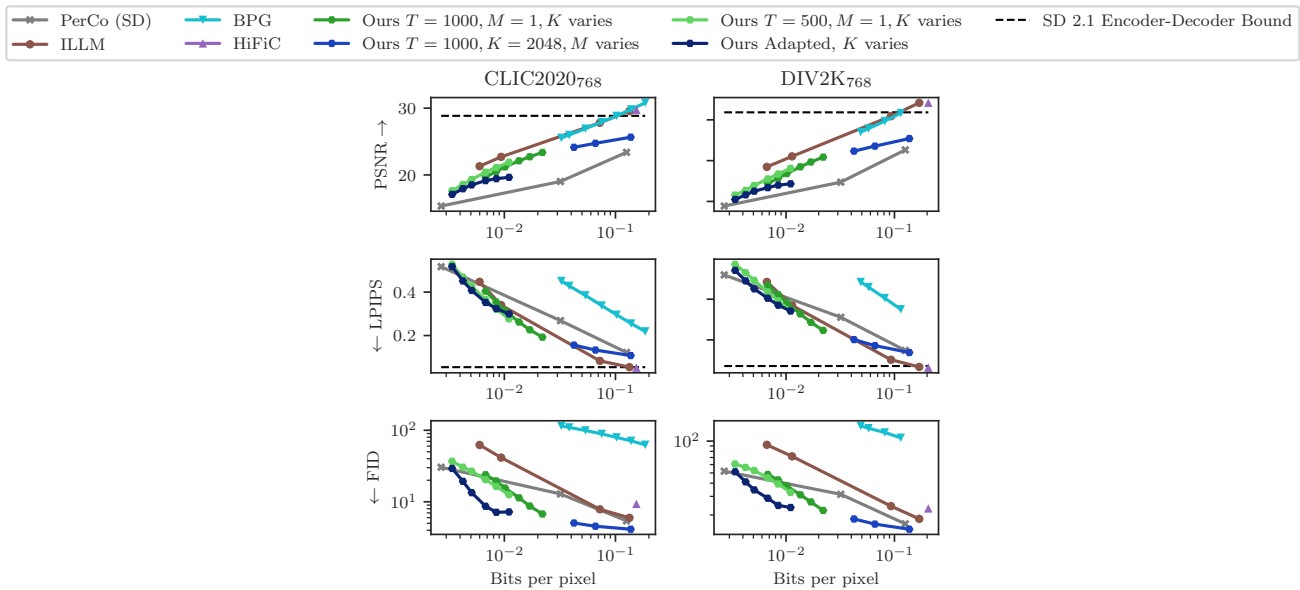

*Figure 15.* **Quantitative image compression results on** $768 \times 768$ **sized images.** At higher bit rates, our method achieves the lowest (best) FID scores in both datasets while maintaining better distortion metrics compared to PerCo (SD). At extremely low bit rates, while PerCo (SD) shows marginally better FID scores, our method attains superior PSNR performance.

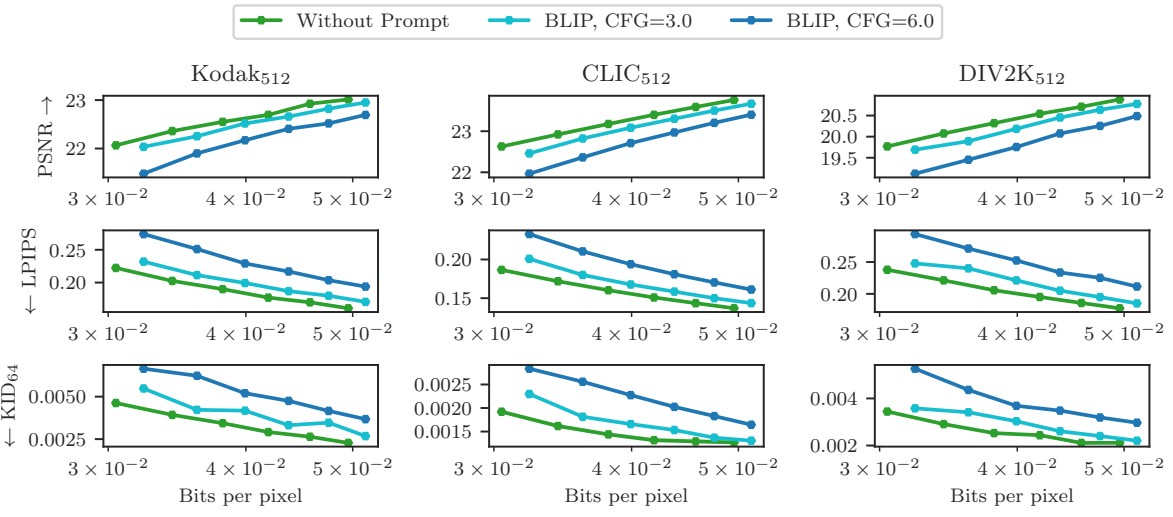

*Figure 16.* **Evaluating the effectiveness of using text prompts in image compression.** We evaluate our unconditional compression method with the text-conditional one, while using the text captions generated by BLIP-2. We find that using such text prompts hinders our compression results, both in terms of perceptual quality and distortion.

*Table 2.* Compression quantitative evaluations, for the ImageNet $256 \times 256$ dataset.

| Method | BPP | FID | LPIPS | PSNR |
|--------|-----|-----|-------|------|
| BPG | 0.056 | 144.322 | 0.460 | 23.771 |
| | 0.118 | 111.821 | 0.330 | 26.085 |
| | 0.248 | 81.555 | 0.218 | 28.329 |
| | 0.503 | 60.512 | 0.127 | 30.864 |
| | 0.941 | 42.714 | 0.064 | 33.602 |
| HiFiC | 0.220 | 42.610 | 0.055 | 27.713 |
| | 0.395 | 31.567 | 0.033 | 29.849 |
| | 0.584 | 25.125 | 0.021 | 31.622 |
| IPIC | 0.095 | 45.367 | 0.211 | 24.470 |
| | 0.148 | 38.413 | 0.152 | 26.011 |
| | 0.228 | 36.301 | 0.132 | 27.664 |
| | 0.550 | 27.075 | 0.074 | 31.178 |
| | 0.945 | 20.689 | 0.044 | 33.800 |
| PSC-D | 0.019 | 147.317 | 0.591 | 21.648 |
| | 0.039 | 127.275 | 0.496 | 23.572 |
| | 0.077 | 105.225 | 0.404 | 25.605 |
| | 0.155 | 86.787 | 0.316 | 27.711 |
| PSC-P | 0.019 | 67.457 | 0.431 | 18.575 |
| | 0.039 | 59.119 | 0.353 | 20.178 |
| | 0.077 | 51.464 | 0.268 | 22.649 |
| | 0.155 | 44.589 | 0.194 | 25.107 |
| Ours | 0.018 | 68.339 | 0.419 | 19.624 |
| | 0.023 | 63.731 | 0.365 | 20.566 |
| | 0.032 | 57.027 | 0.296 | 21.877 |
| | 0.041 | 52.157 | 0.251 | 22.795 |
| | 0.053 | 47.834 | 0.220 | 23.401 |
| | 0.069 | 43.972 | 0.185 | 24.199 |
| | 0.091 | 40.787 | 0.165 | 24.643 |
| | 0.122 | 36.890 | 0.134 | 25.587 |
| | 0.183 | 31.977 | 0.102 | 26.765 |
| | 0.381 | 28.247 | 0.076 | 27.949 |
| | 0.594 | 25.705 | 0.063 | 28.694 |
| | 0.808 | 24.165 | 0.055 | 29.211 |
| | 1.021 | 22.938 | 0.050 | 29.596 |

Table 3. Compression quantitative evaluations, for the Kodak24 $512 \times 512$ dataset.

| Method | BPP | FID | LPIPS | PSNR |
|---|---|---|---|---|
| **BPG** | 0.037 | 148.306 | 0.540 | 24.211 |
| | 0.044 | 141.575 | 0.513 | 24.638 |
| | 0.062 | 128.095 | 0.461 | 25.539 |
| | 0.088 | 114.256 | 0.407 | 26.492 |
| | 0.121 | 104.289 | 0.357 | 27.375 |
| | 0.167 | 92.423 | 0.307 | 28.346 |
| | 0.227 | 82.184 | 0.260 | 29.359 |
| **CRDR-D** | 0.114 | 94.475 | 0.257 | 27.988 |
| | 0.210 | 76.015 | 0.167 | 30.069 |
| **CRDR-R** | 0.114 | 35.676 | 0.094 | 27.315 |
| | 0.210 | 28.391 | 0.057 | 29.373 |
| **HiFiC** | 0.191 | 31.713 | 0.068 | 27.376 |
| | 0.363 | 25.734 | 0.042 | 29.466 |
| **ILLM** | 0.085 | 36.080 | 0.110 | 25.683 |
| | 0.159 | 28.556 | 0.072 | 27.296 |
| | 0.304 | 24.448 | 0.044 | 29.395 |
| **PSC-P** | 0.025 | 47.334 | 0.473 | 18.516 |
| | 0.049 | 40.691 | 0.371 | 20.080 |
| | 0.095 | 35.314 | 0.266 | 21.683 |
| **PerCo (SD)** | 0.033 | 37.019 | 0.307 | 19.017 |
| | 0.127 | 26.418 | 0.145 | 22.325 |
| **Ours** | 0.030 | 32.031 | 0.222 | 22.066 |
| | 0.038 | 29.117 | 0.190 | 22.551 |
| | 0.050 | 25.647 | 0.161 | 23.013 |
| | 0.095 | 24.215 | 0.138 | 23.606 |
| | 0.149 | 23.199 | 0.124 | 24.069 |
| | 0.309 | 22.260 | 0.108 | 24.665 |
| **SD 2.1 Encoder-Decoder Bound** | – | – | 0.071 | 26.428 |

*Table 4.* Compression quantitative evaluations, for the CLIC2020 $512 \times 512$ dataset.

| Method | BPP | FID | LPIPS | PSNR |
|---|---|---|---|---|
| **BPG** | 0.040 | 112.453 | 0.451 | 24.777 |
| | 0.046 | 106.936 | 0.428 | 25.213 |
| | 0.064 | 96.809 | 0.380 | 26.149 |
| | 0.088 | 86.578 | 0.331 | 27.138 |
| | 0.119 | 77.203 | 0.285 | 28.072 |
| | 0.160 | 68.827 | 0.242 | 29.049 |
| | 0.214 | 60.534 | 0.204 | 30.063 |
| **CRDR-D** | 0.106 | 52.459 | 0.193 | 28.934 |
| | 0.188 | 40.228 | 0.124 | 31.062 |
| | 0.355 | 27.793 | 0.070 | 33.705 |
| **CRDR-R** | 0.106 | 14.292 | 0.074 | 28.352 |
| | 0.188 | 10.022 | 0.045 | 30.452 |
| | 0.355 | 6.225 | 0.025 | 33.028 |
| **HiFiC** | 0.173 | 14.000 | 0.053 | 28.878 |
| | 0.322 | 10.157 | 0.032 | 31.040 |
| **ILLM** | 0.081 | 12.666 | 0.087 | 26.898 |
| | 0.146 | 9.923 | 0.056 | 28.662 |
| | 0.271 | 8.246 | 0.034 | 30.840 |
| | 0.391 | 7.222 | 0.024 | 32.233 |
| **PSC-P** | 0.025 | 15.007 | 0.435 | 18.153 |
| | 0.049 | 13.527 | 0.335 | 20.110 |
| | 0.095 | 11.879 | 0.229 | 22.276 |
| **PerCo (SD)** | 0.033 | 13.896 | 0.287 | 18.111 |
| | 0.127 | 7.888 | 0.128 | 22.453 |
| **Ours** | 0.030 | 9.459 | 0.186 | 22.630 |
| | 0.038 | 8.386 | 0.160 | 23.171 |
| | 0.050 | 7.755 | 0.137 | 23.748 |
| | 0.095 | 7.340 | 0.116 | 24.472 |
| | 0.149 | 7.164 | 0.103 | 25.008 |
| | 0.309 | 6.825 | 0.088 | 25.782 |
| **SD 2.1 Encoder-Decoder Bound** | – | – | 0.056 | 27.901 |

*Table 5.* Compression quantitative evaluations, for the DIV2K $512 \times 512$ dataset.

| Method | BPP | FID | LPIPS | PSNR |
|---|---|---|---|---|
| **BPG** | 0.058 | 152.623 | 0.490 | 22.793 |
| | 0.068 | 146.219 | 0.463 | 23.217 |
| | 0.096 | 134.029 | 0.407 | 24.124 |
| | 0.134 | 122.895 | 0.348 | 25.095 |
| | 0.327 | 87.879 | 0.192 | 28.049 |
| **CRDR-D** | 0.152 | 89.766 | 0.203 | 26.618 |
| | 0.274 | 69.466 | 0.118 | 28.762 |
| **CRDR-R** | 0.152 | 39.291 | 0.081 | 26.110 |
| | 0.274 | 31.422 | 0.048 | 28.275 |
| **HiFiC** | 0.226 | 39.832 | 0.066 | 26.138 |
| **ILLM** | 0.104 | 42.591 | 0.107 | 24.579 |
| | 0.187 | 34.103 | 0.069 | 26.206 |
| | 0.332 | 29.267 | 0.043 | 28.229 |
| **PSC-P** | 0.025 | 48.350 | 0.474 | 15.869 |
| | 0.049 | 44.267 | 0.386 | 17.470 |
| | 0.095 | 38.406 | 0.281 | 19.220 |
| **PerCo (SD)** | 0.033 | 40.944 | 0.325 | 16.518 |
| | 0.127 | 29.516 | 0.155 | 20.483 |
| **Ours** | 0.030 | 36.283 | 0.238 | 19.769 |
| | 0.038 | 32.917 | 0.206 | 20.319 |
| | 0.050 | 30.724 | 0.177 | 20.881 |
| | 0.095 | 29.322 | 0.150 | 21.566 |
| | 0.149 | 28.011 | 0.132 | 22.107 |
| | 0.309 | 26.756 | 0.114 | 22.867 |
| **SD 2.1 Encoder-Decoder Bound** | – | – | 0.076 | 24.878 |

*Table 6.* Compression quantitative evaluations, for the CLIC2020 $768 \times 768$ dataset.

| Method | BPP | FID | LPIPS | PSNR |
|---|---|---|---|---|
| **BPG** | 0.032 | 115.009 | 0.451 | 25.547 |
| | 0.038 | 109.587 | 0.429 | 25.984 |
| | 0.054 | 99.482 | 0.385 | 26.925 |
| | 0.075 | 89.013 | 0.338 | 27.915 |
| | 0.102 | 79.456 | 0.296 | 28.840 |
| | 0.138 | 71.107 | 0.255 | 29.811 |
| | 0.185 | 62.617 | 0.219 | 30.812 |
| **HiFiC** | 0.154 | 9.359 | 0.052 | 29.745 |
| **ILLM** | 0.006 | 62.327 | 0.447 | 21.304 |
| | 0.009 | 41.479 | 0.340 | 22.703 |
| | 0.072 | 7.852 | 0.084 | 27.808 |
| | 0.134 | 5.966 | 0.055 | 29.567 |
| **PerCo (SD)** | 0.003 | 30.409 | 0.517 | 15.339 |
| | 0.032 | 12.869 | 0.269 | 19.018 |
| | 0.126 | 5.419 | 0.122 | 23.387 |
| **Ours** | 0.007 | 23.862 | 0.404 | 19.672 |
| | 0.008 | 19.521 | 0.354 | 20.532 |
| | 0.010 | 15.559 | 0.314 | 21.207 |
| | 0.014 | 11.362 | 0.262 | 22.116 |
| | 0.017 | 8.722 | 0.227 | 22.722 |
| | 0.022 | 6.753 | 0.192 | 23.366 |
| | 0.042 | 5.051 | 0.156 | 24.136 |
| | 0.066 | 4.549 | 0.133 | 24.739 |
| | 0.137 | 4.132 | 0.108 | 25.650 |
| **SD 2.1 Encoder-Decoder Bound** | – | – | 0.055 | 28.850 |

*Table 7.* Compression quantitative evaluations, for the DIV2K $768 \times 768$ dataset.

| Method | BPP | FID | LPIPS | PSNR |
|---|---|---|---|---|
| **BPG** | 0.049 | 139.204 | 0.485 | 23.504 |
|  | 0.058 | 132.393 | 0.460 | 23.928 |
|  | 0.081 | 120.682 | 0.406 | 24.841 |
|  | 0.114 | 107.472 | 0.350 | 25.826 |
| **HiFiC** | 0.205 | 23.077 | 0.063 | 27.096 |
| **ILLM** | 0.007 | 92.242 | 0.485 | 19.234 |
|  | 0.011 | 71.777 | 0.369 | 20.522 |
|  | 0.093 | 24.253 | 0.102 | 25.446 |
|  | 0.169 | 18.386 | 0.067 | 27.085 |
| **PerCo (SD)** | 0.003 | 51.915 | 0.519 | 14.436 |
|  | 0.032 | 31.378 | 0.311 | 17.364 |
|  | 0.126 | 16.487 | 0.148 | 21.317 |
| **Ours** | 0.007 | 48.315 | 0.469 | 17.194 |
|  | 0.008 | 42.932 | 0.422 | 17.858 |
|  | 0.010 | 37.680 | 0.383 | 18.413 |
|  | 0.014 | 31.038 | 0.328 | 19.246 |
|  | 0.017 | 26.599 | 0.286 | 19.832 |
|  | 0.022 | 22.073 | 0.247 | 20.432 |
|  | 0.042 | 18.330 | 0.201 | 21.167 |
|  | 0.066 | 16.396 | 0.172 | 21.780 |
|  | 0.137 | 14.662 | 0.138 | 22.707 |
| **SD 2.1 Encoder-Decoder Bound** | – | – | 0.071 | 25.919 |

# C. Compressed Conditional Generation Supplementary

## C.1. Background and Proof of Proposition 6.1

We will prove that Proposition 6.1 holds for any score-based diffusion model (Song et al., 2020). For completeness, we first provide the necessary mathematical background and then proceed to the proof of the proposition.

### C.1.1. BACKGROUND

**Score-Based Generative Models.** Score-based generative models (Song et al., 2020) define a diffusion process $\{\mathbf{x}(t) : t \in [0, T]\}$, where $p_0$ and $p_T$ denote the data distribution and the prior distribution, respectively, and $p_t$ denotes the distribution of $x(t)$. Such a diffusion process can generally be modeled as the stochastic differential equation (SDE)

$$d\mathbf{x} = f(\mathbf{x}, t)dt + g(t)d\mathbf{w}, \tag{16}$$

where $f(\cdot, t)$ is called the *drift* coefficient, $g(t)$ is called the *diffusion* coefficient, $\mathbf{w}(t)$ is a standard Wiener process, and $dt$ denotes an infinitesimal timestep. Samples from the data distribution $p_0$ can be generated by solving the reverse-time SDE (Anderson, 1982),

$$d\mathbf{x} = \left[ f(\mathbf{x}, t) - g^2(t)\boldsymbol{s}_t(\mathbf{x}) \right] dt + g(t)d\bar{\mathbf{w}}, \tag{17}$$

starting from samples of $\mathbf{x}(T)$. Here, $\boldsymbol{s}_t(\mathbf{x}(t)) \coloneqq \nabla_{\mathbf{x}(t)} \log p_t(\mathbf{x}(t))$ is the *score* of $p_t$, $\bar{\mathbf{w}}(t)$ denotes a standard Wiener process where time flows backwards, and $dt$ is an infinitesimal *negative* timestep. Samples from the data distribution can also be generated by solving the *probability flow* ODE,

$$d\mathbf{x} = \left[ f(\mathbf{x}, t) - \frac{1}{2}g^2(t)\boldsymbol{s}_t(\mathbf{x}) \right] dt. \tag{18}$$

**Solving The Reverse-Time SDE.** The reverse-time SDE in Eq. (17) can be solved with any numerical SDE solver (e.g., Euler-Maruyama), which corresponds to some time discretization of the forward and reverse stochastic dynamics. For the sake of our proof, we adopt the simple solver proposed by Song et al. (2020),

$$\mathbf{x}_{i-1} = \mathbf{x}_i - f_i(\mathbf{x}_i) + g_i^2 \mathbf{s}_i(\mathbf{x}_i) + g_i \mathbf{z}_i, \quad \mathbf{z}_i \sim \mathcal{N}(\mathbf{0}, \boldsymbol{I}), \tag{19}$$

where $i = T, \dots, 1$ and $f_i$ and $g_i$ are the time-discretized versions of $f$ and $g$, respectively. Note that DDPMs (Ho et al., 2020) are score-based diffusion models that solve a reverse-time Variance Preserving (VP) SDE, where $f(\mathbf{x}(t), t) = -\frac{1}{2}\beta(t)\mathbf{x}(t)$ and $g(t) = \sqrt{\beta(t)}$ for some function $\beta$.

### C.1.2. PROOF OF PROPOSITION 6.1

Given any general score-based diffusion model, we can write the DDCM compressed conditional generation process as

$$\mathbf{x}_{i-1} = \mathbf{x}_i - f_i(\mathbf{x}_i) + g_i^2 \mathbf{s}_i(\mathbf{x}_i) + g_i \mathcal{C}_i(k_i), \tag{20}$$

where $k_i$ are picked according to Eq. (8). Choosing $\mathcal{L} = \mathcal{L}_{\mathrm{P}}$, we have

$$k_i = \underset{k \in \{1, \dots, K\}}{\arg\min} \|\mathcal{C}_i(k) - g_i \nabla_{\mathbf{x}_i} \log p_i(\mathbf{y}|\mathbf{x}_i)\|^2. \tag{21}$$

Since each $\mathcal{C}_i$ contains $K$ independent samples drawn from a normal distribution $\mathcal{N}(\mathbf{0}, \boldsymbol{I})$, we have

$$\{\mathcal{C}_i(1), \dots, \mathcal{C}_i(K)\} \underset{K \to \infty}{\longrightarrow} \mathbb{R}^n, \tag{22}$$

where $n$ denotes the dimensionality of each vector in $\mathcal{C}_i$, and $\{\mathcal{C}_i(1), \dots, \mathcal{C}_i(K)\}$ is the set comprised of all the elements in the $\mathcal{C}_i$ (without repetition). Since $g_i \nabla_{\mathbf{x}_i} \log p_i(\mathbf{y}|\mathbf{x}_i) \in \mathbb{R}^n$, we have

$$\min_{k \in \{1, \dots, K\}} \|\mathcal{C}_i(k) - g_i \nabla_{\mathbf{x}_i} \log p_i(\mathbf{y}|\mathbf{x}_i)\|^2 \underset{K \to \infty}{\longrightarrow} 0. \tag{23}$$

Thus,

$$\mathcal{C}_i(k_i) \xrightarrow[K \to \infty]{} g_i \nabla_{\mathbf{x}_i} \log p_i(\mathbf{y}|\mathbf{x}_i). \tag{24}$$

Plugging Eq. (24) into Eq. (20), we get

$$\mathbf{x}_{i-1} \xrightarrow[K \to \infty]{} \mathbf{x}_i - f_i(\mathbf{x}_i) + g_i^2 \mathbf{s}_i(\mathbf{x}_i) + g_i^2 \nabla_{\mathbf{x}_i} \log p_i(\mathbf{y}|\mathbf{x}_i) \tag{25}$$

$$= \mathbf{x}_i - f_i(\mathbf{x}_i) + g_i^2 \nabla_{\mathbf{x}_i} \log p_i(\mathbf{x}_i) + g_i^2 \nabla_{\mathbf{x}_i} \log p_i(\mathbf{y}|\mathbf{x}_i) \tag{26}$$

$$= \mathbf{x}_i - f_i(\mathbf{x}_i) + g_i^2 [\nabla_{\mathbf{x}_i} \log p_i(\mathbf{x}_i) + \nabla_{\mathbf{x}_i} \log p_i(\mathbf{y}|\mathbf{x}_i)] \tag{27}$$

$$= \mathbf{x}_i - f_i(\mathbf{x}_i) + g_i^2 \nabla_{\mathbf{x}_i} \log p_i(\mathbf{x}_i|\mathbf{y}), \tag{28}$$

where in Eq. (28) we used Bayes rule and the fact that $\nabla_{\mathbf{x}_i} \log p(\mathbf{y}) = 0$. Note that Eq. (28) resembles a time discretization of a probability flow ODE (Eq. (18)) over the posterior distribution $p_0(\mathbf{x}_0|\mathbf{y})$, with $f(\cdot, t)$ and $\sqrt{2}g(t)$ being the drift and diffusion coefficients in continuous time, respectively. Thus, when $K \to \infty$, our compressed conditional generation process becomes a sampler from $p_0(\mathbf{x}_0|\mathbf{y})$.

## C.2. Image Compression as a Private Case of Compressed Conditional Generation

We show that our standard image compression scheme from Sec. 5 is a private case of our compressed conditional generation scheme from Sec. 6, where $\mathbf{y} = \mathbf{x}_0$ and $\mathcal{L} = \mathcal{L}_P$. When $\mathbf{y} = \mathbf{x}_0$ we have

$$\nabla_{\mathbf{x}_i} \log p_i(\mathbf{y}|\mathbf{x}_i) = \nabla_{\mathbf{x}_i} \log p_i(\mathbf{x}_0|\mathbf{x}_i)$$

$$= \nabla_{\mathbf{x}_i} \log p_i(\mathbf{x}_i|\mathbf{x}_0) - \nabla_{\mathbf{x}_i} \log p_i(\mathbf{x}_i) \tag{29}$$

$$= \nabla_{\mathbf{x}_i} \log p_i(\mathbf{x}_i|\mathbf{x}_0) - \mathbf{s}_i(\mathbf{x}_i) \tag{30}$$

where $\nabla_{\mathbf{x}_i} \log p_i(\mathbf{x}_i|\mathbf{x}_0)$ can be computed in closed-form via the forward diffusion process in Eq. (1). In particular, we have $p_i(\mathbf{x}_i|\mathbf{x}_0) = \mathcal{N}(\mathbf{x}_i; \sqrt{\bar{\alpha}_i}\mathbf{x}_0, (1 - \bar{\alpha}_i)\mathbf{I})$ (Ho et al., 2020), and thus

$$\nabla_{\mathbf{x}_i} \log p_i(\mathbf{x}_i|\mathbf{x}_0) = -\nabla_{\mathbf{x}_i} \frac{\|\mathbf{x}_i - \sqrt{\bar{\alpha}_i}\mathbf{x}_0\|^2}{2(1 - \bar{\alpha}_i)} \tag{31}$$

$$= -\frac{\mathbf{x}_i - \sqrt{\bar{\alpha}_i}\mathbf{x}_0}{1 - \bar{\alpha}_i} \tag{32}$$

$$= \frac{\sqrt{\bar{\alpha}_i}\mathbf{x}_0 - \mathbf{x}_i}{1 - \bar{\alpha}_i}. \tag{33}$$

Moreover, it is well known that (Ho et al., 2020; Song et al., 2020)

$$\mathbf{s}_i(\mathbf{x}_i) = \frac{\sqrt{\bar{\alpha}_i}\hat{\mathbf{x}}_{0|i} - \mathbf{x}_i}{1 - \bar{\alpha}_i}, \tag{34}$$

where $\hat{\mathbf{x}}_{0|i}$ is the (time-aware) Minimum Mean-Squared-Error (MMSE) estimator of $\mathbf{x}_0$ given $\mathbf{x}_i$. Plugging Eqs. (33) and (34) into (30), we get

$$\nabla_{\mathbf{x}_i} \log p_i(\mathbf{x}_0|\mathbf{x}_i) = \frac{\sqrt{\bar{\alpha}_i}}{1 - \bar{\alpha}_i}(\mathbf{x}_0 - \hat{\mathbf{x}}_{0|i}). \tag{35}$$

Thus, we have

$$k_i = \underset{k \in \{1, \ldots, K\}}{\arg\min} \mathcal{L}_P(\mathbf{y}, \mathbf{x}_i, \mathcal{C}_i, k) \tag{36}$$

$$= \underset{k \in \{1, \ldots, K\}}{\arg\min} \left\| \mathcal{C}_i(k) - \sigma_i \frac{\sqrt{\bar{\alpha}_i}}{1 - \bar{\alpha}_i}(\mathbf{x}_0 - \hat{\mathbf{x}}_{0|i}) \right\|^2 \tag{37}$$

$$= \underset{k \in \{1, \ldots, K\}}{\arg\min} \|\mathcal{C}_i(k)\|^2 - 2\langle \mathcal{C}_i(k), \sigma_i \frac{\sqrt{\bar{\alpha}_i}}{1 - \bar{\alpha}_i}(\mathbf{x}_0 - \hat{\mathbf{x}}_{0|i})\rangle + \left\| \sigma_i \frac{\sqrt{\bar{\alpha}_i}}{1 - \bar{\alpha}_i}(\mathbf{x}_0 - \hat{\mathbf{x}}_{0|i}) \right\|^2 \tag{38}$$

$$= \underset{k \in \{1, \ldots, K\}}{\arg\min} \|\mathcal{C}_i(k)\|^2 - 2\langle \mathcal{C}_i(k), \sigma_i \frac{\sqrt{\bar{\alpha}_i}}{1 - \bar{\alpha}_i}(\mathbf{x}_0 - \hat{\mathbf{x}}_{0|i})\rangle. \tag{39}$$

Below, we show that $\|\mathcal{C}_i(k)\|^2$ is roughly equal for every $k$. Thus, it holds that

$$k_i = \underset{k \in \{1,\ldots,K\}}{\arg\min} \ \|\mathcal{C}_i(k)\|^2 - 2\langle \mathcal{C}_i(k), \sigma_i \frac{\sqrt{\bar{\alpha}_i}}{1 - \bar{\alpha}_i}(\mathbf{x}_0 - \hat{\mathbf{x}}_{0|i})\rangle \tag{40}$$

$$\approx \underset{k \in \{1,\ldots,K\}}{\arg\min} \ \text{const} - 2\langle \mathcal{C}_i(k), \sigma_i \frac{\sqrt{\bar{\alpha}_i}}{1 - \bar{\alpha}_i}(\mathbf{x}_0 - \hat{\mathbf{x}}_{0|i})\rangle \tag{41}$$

$$= \underset{k \in \{1,\ldots,K\}}{\arg\max} \ \langle \mathcal{C}_i(k), \sigma_i \frac{\sqrt{\bar{\alpha}_i}}{1 - \bar{\alpha}_i}(\mathbf{x}_0 - \hat{\mathbf{x}}_{0|i})\rangle \tag{42}$$

$$= \underset{k \in \{1,\ldots,K\}}{\arg\max} \ \sigma_i \frac{\sqrt{\bar{\alpha}_i}}{1 - \bar{\alpha}_i} \langle \mathcal{C}_i(k), \mathbf{x}_0 - \hat{\mathbf{x}}_{0|i}\rangle \tag{43}$$

$$= \underset{k \in \{1,\ldots,K\}}{\arg\max} \ \langle \mathcal{C}_i(k), \mathbf{x}_0 - \hat{\mathbf{x}}_{0|i}\rangle. \tag{44}$$

Note that the noise selection strategy in Eq. (44) is similar to that of our standard compression scheme, namely Eq. (7). Thus, our compression method is a private case of our compressed conditional generation approach. In practice, we used Eq. (7) instead of Eq. (36) since the former worked slightly better.

To show that $\|\mathcal{C}_i(k)\|^2$ is roughly constant for every $k$, note that $\mathcal{C}_i(k)$ is a sample from a $n$-dimensional multivariate normal distribution $\mathcal{N}(\mathbf{0}, \boldsymbol{I})$. Thus, $\|\mathcal{C}_i(k)\|^2$ is a sample from a chi-squared distribution with $n$ degrees of freedom. It is well known that samples from this distribution strongly concentrate around its mean $n$ for large values of $n$. Namely, $\|\mathcal{C}_i(k)\|^2$ is highly likely to be close to $n$, especially for relatively small values of $K$.

| Original | Degraded | DPS | Compr. DPS | DDNM | Compr. DDNM | **Ours** |
| --- | --- | --- | --- | --- | --- | --- |

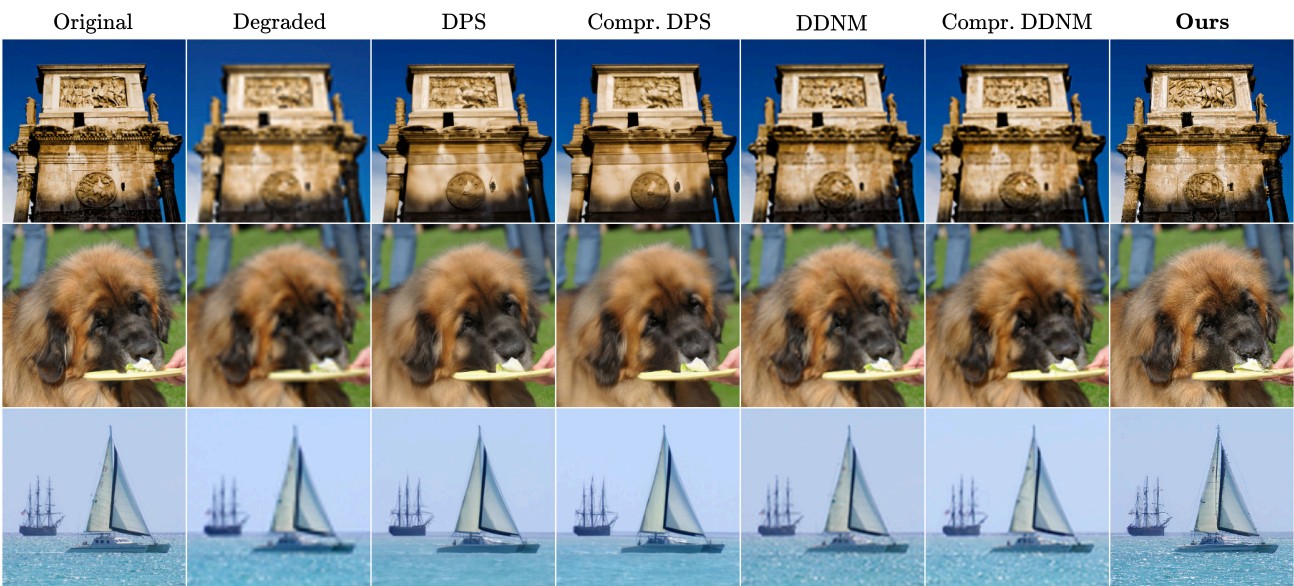

*Figure 17.* **Qualitative comparison of zero-shot image super-resolution methods (posterior sampling).** Our approach clearly produces better output perceptual quality compared to previous methods.

### C.3. Compressed Posterior Sampling for Image Restoration

DPS and DDNM are implemented with the official settings recommended by the authors (Chung et al., 2023; Wang et al., 2023b). Specifically, DPS uses DDPM with $T = 1000$ sampling steps, and DDNM uses DDIM with $\eta = 0.85$ and $T = 100$ sampling steps. We also tried $T = 1000$ for DDNM and found that $T = 100$ works slightly better for the tasks considered. The additional qualitative comparisons in Figs. 17 and 18 further demonstrate that our method produces better output perceptual quality compared to DPS and DDNM.

| Original | Degraded | DPS | Compr. DPS | DDNM | Compr. DDNM | **Ours** |
|----------|----------|-----|------------|------|-------------|----------|

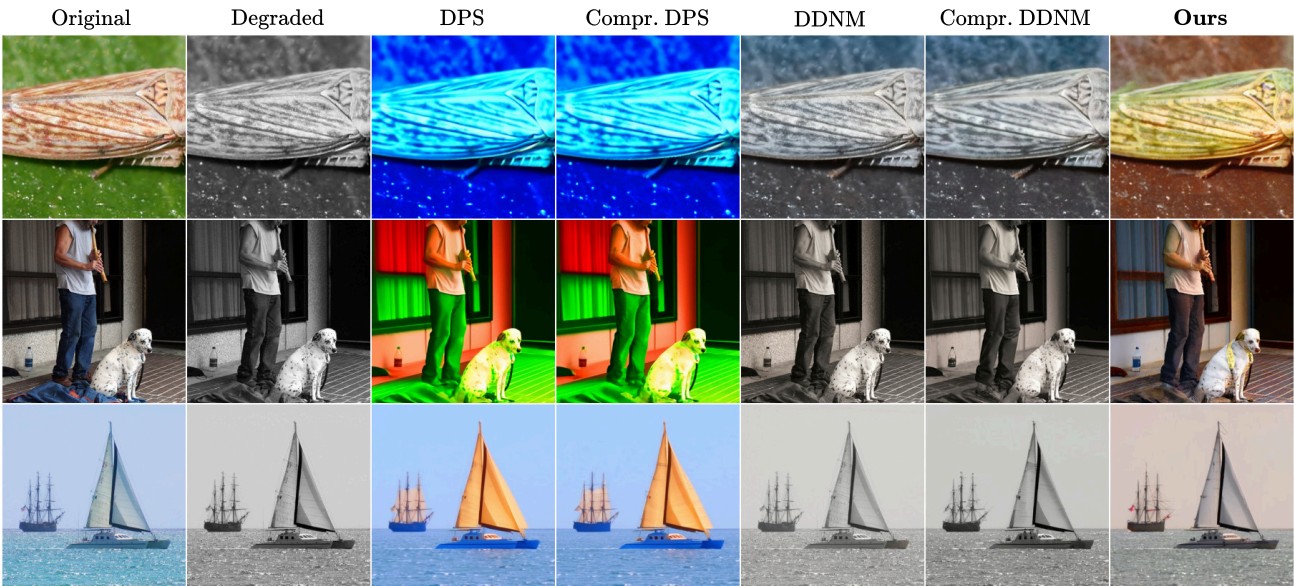

*Figure 18.* **Qualitative comparison of zero-shot image colorization methods (posterior sampling).** Our approach clearly produces better output perceptual quality compared to previous methods.

## C.4. Compressed Real-World Face Image Restoration

### C.4.1. EXPLAINING THE CHOICE OF $r(\mathbf{y})$

To explain our choice of $r(\mathbf{y}) \approx \mathbb{E}[\mathbf{x}_0|\mathbf{y}]$, first note that the MSE of *any* estimator $\hat{\mathbf{x}}_0$ of $\mathbf{x}_0$ given $\mathbf{y}$ can be written as (Freirich et al., 2021)

$$\text{MSE}(\mathbf{x}_0, \hat{\mathbf{x}}_0) = \text{MSE}(\mathbf{x}_0, r(\mathbf{y})) + \text{MSE}(r(\mathbf{y}), \hat{\mathbf{x}}_0)$$
$$= \text{MSE}(r(\mathbf{y}), \hat{\mathbf{x}}_0) + c. \tag{45}$$

where $c$, the MMSE, does not depend on $\hat{\mathbf{x}}_0$. In theory, our goal is to optimize the tradeoff between the MSE of $\hat{\mathbf{x}}_0$ and its output perceptual quality according to some quality measure $Q$. This can be accomplished by solving

$$\min_{\hat{\mathbf{x}}_0} \text{MSE}(\mathbf{x}_0, \hat{\mathbf{x}}_0) + \lambda Q(\hat{\mathbf{x}}_0), \tag{46}$$

where $\lambda$ is some hyper-parameter that controls the perception-distortion tradeoff. At test time, however, we do not have access to the original image $\mathbf{x}_0$. By plugging Eq. (45) into Eq. (46) we obtain

$$\min_{\hat{\mathbf{x}}_0} \text{MSE}(\mathbf{x}_0, \hat{\mathbf{x}}_0) + \lambda Q(\hat{\mathbf{x}}_0) = \min_{\hat{\mathbf{x}}_0} \text{MSE}(r(\mathbf{y}), \hat{\mathbf{x}}_0) + c + \lambda Q(\hat{\mathbf{x}}_0)$$
$$= \min_{\hat{\mathbf{x}}_0} \text{MSE}(r(\mathbf{y}), \hat{\mathbf{x}}_0) + \lambda Q(\hat{\mathbf{x}}_0). \tag{47}$$

Namely, as long as $r(\mathbf{y})$ is a good approximation of the *true* MMSE estimator, we can rely on it for optimizing tradeoff (46) without having access to $\mathbf{x}_0$. This resembles our approach in Sec. 6.2, where we *greedily* optimize Eq. (47) throughout the trajectory of the DDCM sampling process.

### C.4.2. ADDITIONAL DETAILS AND EXPERIMENTS

We use the PyIQA package to compute all perceptual quality measures, with `niqe` for NIQE, `topiq_nr-face` for TOPIQ, `clipiqa+` for CLIP-IQA$^+$, and `fid_dinov2` for $\text{FD}_{\text{DINOv2}}$, adopting the default settings for each measure. Additional qualitative and quantitative comparisons are shown in Figs. 19, 20, 21 and 22.

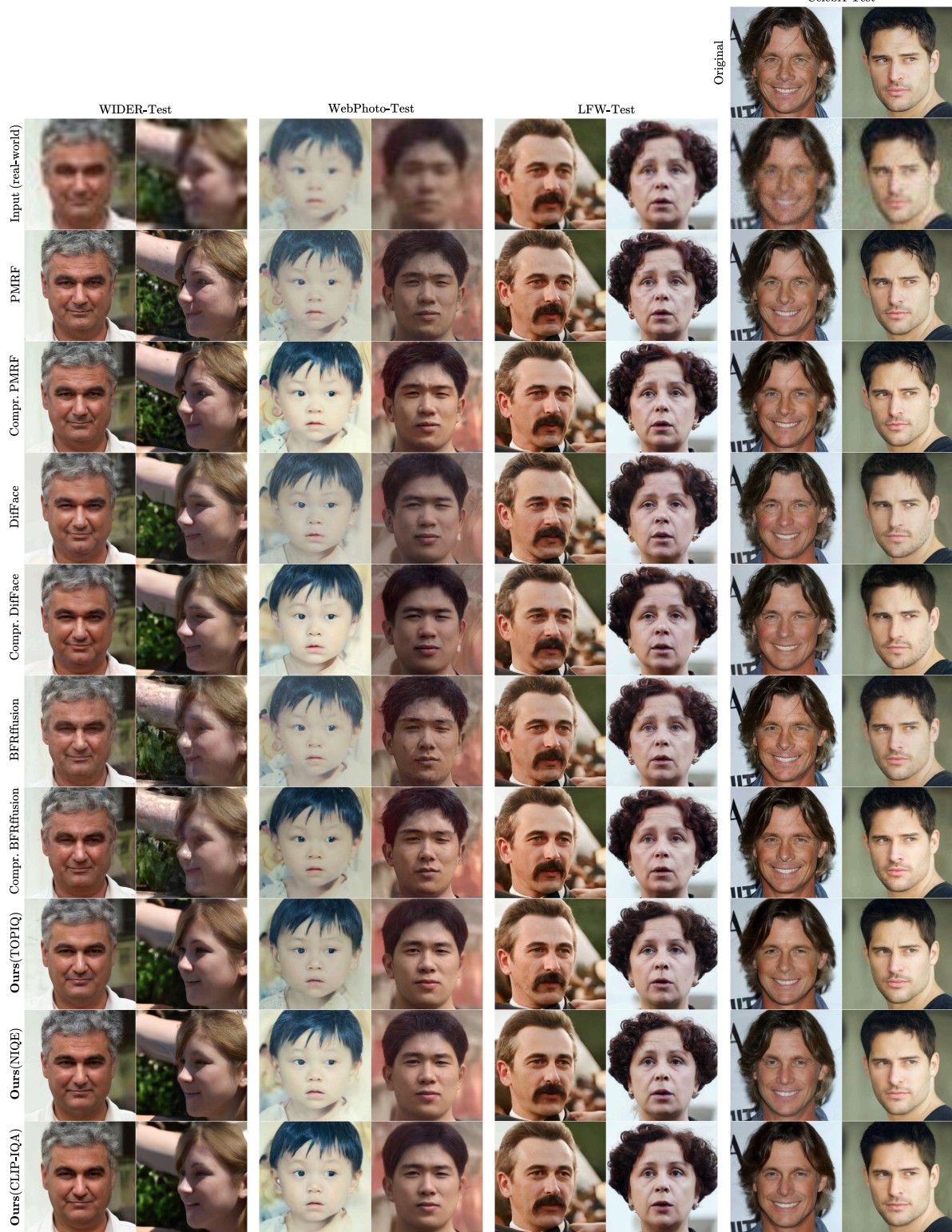

*Figure 19.* **Qualitative comparison of real-world face image restoration methods**. Our method produces high perceptual quality results with less artifacts compared to previous methods, especially for challenging datasets such as WIDER-Test.

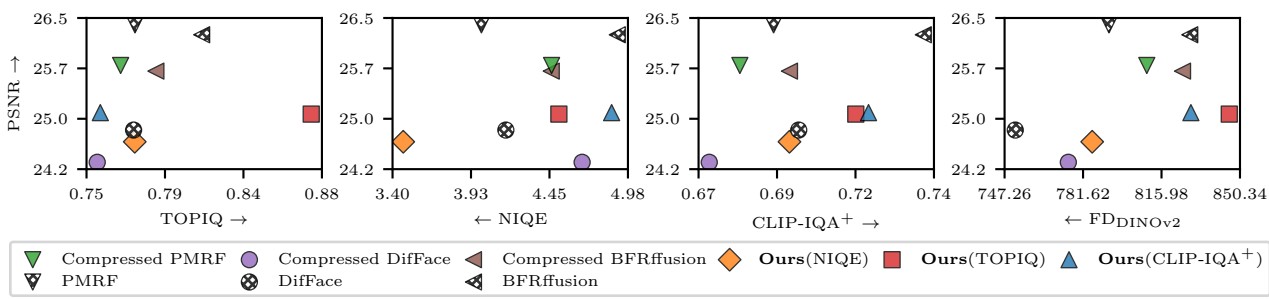

*Figure 20.* **Quantitative comparison of real-world face image restoration methods, evaluated on the CelebA-Test dataset.** We successfully optimize each NR-IQA measure, surpassing the scores of previous methods. Here, only our NIQE-based solution generalizes well to FD$_{\text{DINOv2}}$ in terms of perceptual quality.

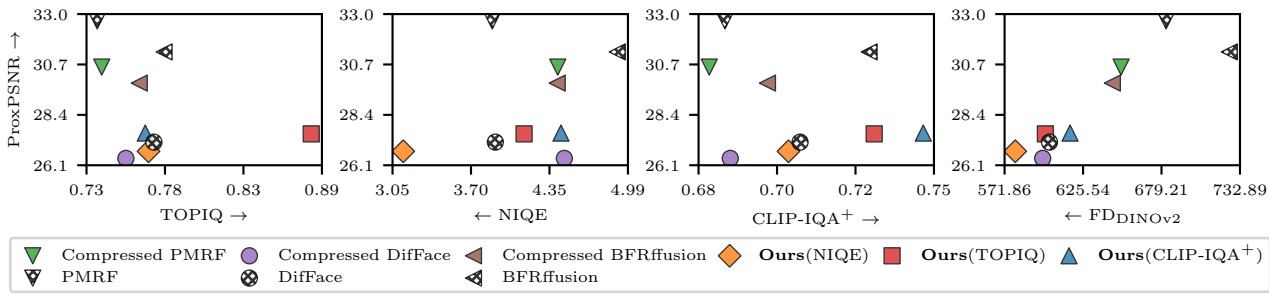

*Figure 21.* **Quantitative comparison of real-world face image restoration methods, evaluated on the LFW-Test dataset.** We successfully optimize each NR-IQA measure, surpassing the scores of previous methods. All our solutions achieve impressive FD$_{\text{DINOv2}}$ scores, while our NIQE-based solution surpasses all methods according to this measure.

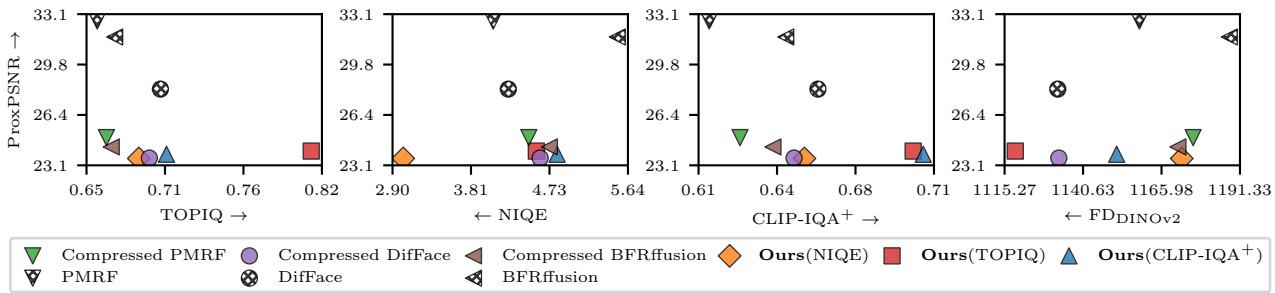

*Figure 22.* **Quantitative comparison of real-world face image restoration methods, evaluated on the WebPhoto-Test dataset.** We successfully optimize each NR-IQA measure, surpassing the scores of previous methods. Our TOPIQ-based solution achieves the best FD$_{\text{DINOv2}}$ scores compared to all methods..

## C.5. Compressed Classifier Guidance

Consider the case where $\mathbf{y}$ represents the *class* of an image $\mathbf{x}_0$. An unconditional score-based generative model can be *guided* to generate samples the posterior $p_0(\mathbf{x}_0|\mathbf{y})$, by perturbing the generated samples according to the gradient of a time-dependent trained classifier $c_\theta(\mathbf{y}; \mathbf{x}_i, i) \approx p_i(\mathbf{y}|\mathbf{x}_i)$ (Dhariwal & Nichol, 2021). This approach is known as classifier guidance (CG). Such a guidance method can be interpreted as an attempt to *confuse* the classifier by perturbing its input adversarially (Ho & Salimans, 2021). However, trained classifiers are typically not robust to adversarial perturbations, making their gradients largely unreliable and unaligned with human perception (Szegedy et al., 2014; Tsipras et al., 2019; Ganz et al., 2023). Thus, the standard CG approach has not seen major success (Ho & Salimans, 2021).

We propose an alternative to this method, circumventing the reliance on the classifier's gradient. Specifically, we set $\mathcal{L}$ in Eq. (8) as

$$\mathcal{L}(\mathbf{y}, \mathbf{x}_i, \mathcal{C}_i, k) = -\log c_\theta(\mathbf{y}; \boldsymbol{\mu}_i(\mathbf{x}_i) + \sigma_i \mathcal{C}_i(k), i). \tag{48}$$

Thus, $\mathcal{L}(\mathbf{y}, \mathbf{x}_i, \mathcal{C}_i, k)$ attains a lower value when $\sigma_i \mathcal{C}_i(k)$ points in some direction that maximizes the probability of the class $\mathbf{y}$. Note that since the codebooks remain fixed, choosing $k_i$ (out of $1, \ldots, K$) to minimize Eq. (8) would always lead to the same generated sample for every $\mathbf{y}$. Thus, we promote sample diversity by first *randomly* selecting a subset of $\tilde{K} < K$ indices $k_{i,1}, \ldots, k_{i,\tilde{K}} \sim \mathrm{Unif}(\{1, \ldots, K\})$, and then choosing

$$k_i = \underset{k \in \{k_{i,1}, \ldots, k_{i,\tilde{K}}\}}{\arg\min} \mathcal{L}(\mathbf{y}, \mathbf{x}_i, \mathcal{C}_i, k). \tag{49}$$

We coin our method Compressed CG (CCG).

We compare our proposed CCG with the standard CG using the same unconditional diffusion model and time-dependent classifier trained on ImageNet $256 \times 256$ (Deng et al., 2009; Dhariwal & Nichol, 2021). We compare the methods "on the same grounds" by using the same standard DDPM noise schedule and $T = 1000$ diffusion steps. Our method is assessed with $K = 256$ and $\tilde{K} = 2$, while the standard CG is assessed with CG scales $s \in \{1, 10, 20\}$. The quantitative comparison in Tab. 8 shows that CCG achieves better (lower) FID and FD$_{\mathrm{DINOv2}}$ scores. A visual comparison is provided in Fig. 23. Note that while using DDCM with standard CG does still produces compressed output images, decoding the produced bit-streams requires access to $\mathbf{y}$. Using DDCM with CCG instead sidesteps this limitation, as $\mathbf{y}$ is not needed for decompression.

*Table 8.* Quantitative comparison of *compressed* classifier guidance (CCG) and standard classifier guidance (CG) for ImageNet $256 \times 256$ conditional image generation, using an unconditional DDM and a classifier for guidance. Our proposed CCG not only outperforms CG in terms of generation performance, but also automatically produces compressed image representations.

|  | Compressed CG (**Ours**) $K = 256, \tilde{K} = 2$ | Standard CG $s = 1 \mid 10 \mid 20$ |
|---|---|---|
| FID | **13.669** | $31.548 \mid 14.481 \mid 14.921$ |
| FD$_{\mathrm{DINOv2}}$ | **204.693** | $459.42 \mid 255.41 \mid 248.34$ |

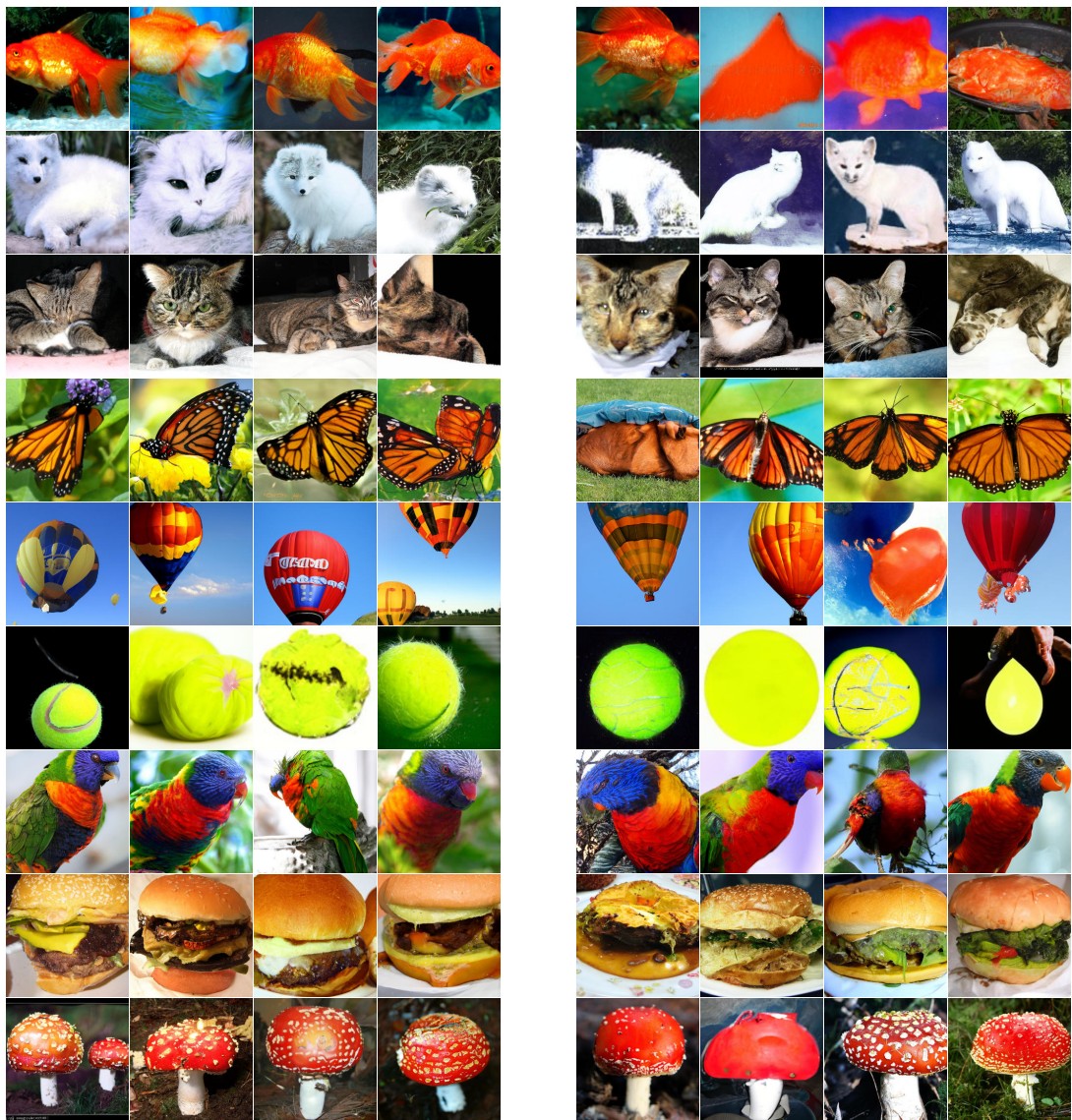

*Figure 23.* **Qualitative comparison of CCG (left) with CG (right).** CCG achieves superior image quality compared to CG while avoiding the use of classifier gradients. Additionally, CCG enables decompression without requiring access to the original class labels.

a male in a black shirt a cellphone and a bridge

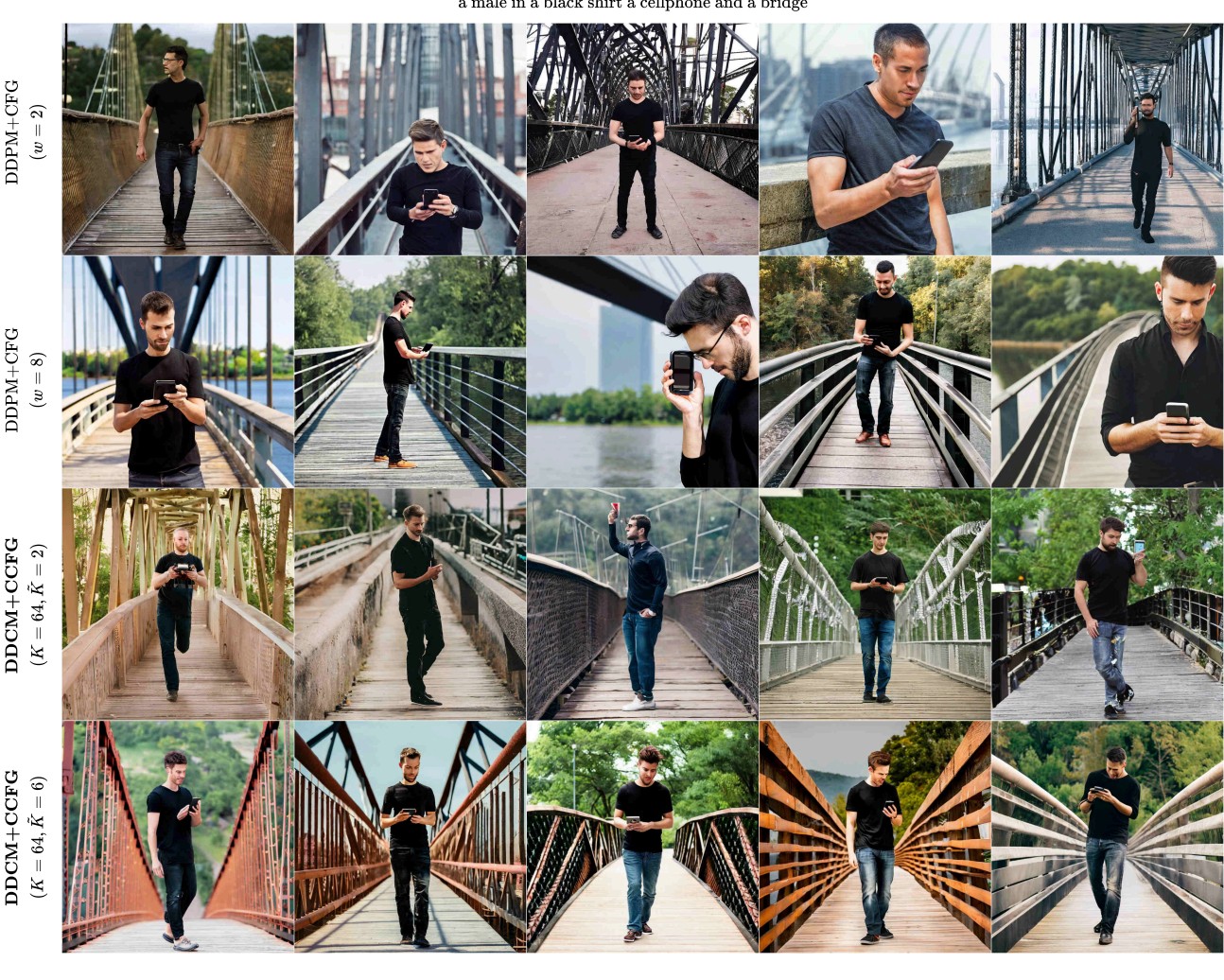

*Figure 24.* **Qualitative comparison of CCFG with CFG.** CCFG achieves comparable image quality and diversity to CFG, while enabling decompression without requiring the original inputs.

## C.6. Compressed Classifier-Free Guidance

The task of text-conditional image generation can be solved using a *conditional* diffusion model, which, theoretically speaking, learns to sample from the posterior distribution $p_0(\mathbf{x}_0|\mathbf{y})$. In practice, however, using a conditional model directly typically yields low fidelity to the inputs. To address this limitation, CG can be used to improve this fidelity at the expense of sample quality and diversity (Dhariwal & Nichol, 2021). Classifier-Free Guidance (CFG) is used more often in practice, as it achieves the same tradeoff by mixing the conditional and unconditional scores during sampling (Ho & Salimans, 2021), thus eliminating the need for a classifier. Particularly, assuming we have access to both the conditional score $\boldsymbol{s}_i(\mathbf{x}_i, \mathbf{y}) \coloneqq \nabla_{\mathbf{x}_i} \log p_i(\mathbf{x}_i|\mathbf{y})$ and the unconditional one $\boldsymbol{s}_i(\mathbf{x}_i)$, CFG proposes to modify the conditional score by

$$\tilde{\boldsymbol{s}}_i(\mathbf{x}_i, \mathbf{y}) = (1 + w)\boldsymbol{s}_i(\mathbf{x}_i, \mathbf{y}) - w\boldsymbol{s}_i(\mathbf{x}_i), \tag{50}$$

where $w$, the CFG scale, is a hyper-parameter controlling the tradeoff between sample quality and diversity.

Here, we introduce a new CFG method that allows generating compressed conditional samples using any pair of conditional and unconditional diffusion models, while controlling the tradeoff between generation quality and the fidelity to the inputs.

The flamingos are outside in the water together.

*Figure 25.* **Qualitative comparison of CCFG with CFG.** CCFG achieves comparable image quality and diversity to CFG, while enabling decompression without requiring the original inputs.

Specifically, since $\nabla_{\mathbf{x}_i} \log p_i(\mathbf{y}|\mathbf{x}_i) = \boldsymbol{s}_i(\mathbf{x}_i|\mathbf{y}) - \boldsymbol{s}_i(\mathbf{x}_i)$, we simply use

$$\mathcal{L}(\mathbf{y}, \mathbf{x}_i, \mathcal{C}_i, k) = -\langle \mathcal{C}_i(k), \boldsymbol{s}_i(\mathbf{x}_i|\mathbf{y}) - \boldsymbol{s}_i(\mathbf{x}_i) \rangle. \tag{51}$$

Note that optimizing Eq. (51) is roughly equivalent to optimizing $\mathcal{L}_\text{P}$ when $\mathbf{x}_i$ is high dimensional (see App. C.2). As in App. C.5, we promote sample diversity by choosing $k_i$ from a randomly sampled subset of $\tilde{K} < K$ indices at each step during the generation. We coin our method Compressed CFG (CCFG).

We implement our method using SD 2.1 trained on $768 \times 768$ images, adopting a DDPM noise schedule with $T = 1000$ diffusion steps, $K = 64$ fixed vectors in each codebook and $\tilde{K} \in \{2, 3, 4, 6, 9\}$. We compare against the same diffusion model with standard DDPM sampling, using $T = 1000$ steps and CFG scales $w \in \{2, 5, 8, 11\}$. The generative performance of both methods is assessed by computing the FID between 10k generated samples and MS-COCO, similarly to Sec. 4. Additionally, we evaluate the alignment between the outputs and the input text prompts using the CLIP score (Hessel et al., 2021) with the OpenAI CLIP ViT-L/14 model (Radford et al., 2021).

Figure 26 shows that our CCFG method is on par with CFG in terms of FID, while CFG produces higher CLIP scores. This suggests that the outputs of CFG better align with the input text prompts compared to CCFG. Yet, the qualitative comparisons in Figs. 24 and 25 show that there is no significant difference between the methods. Importantly, decoding the

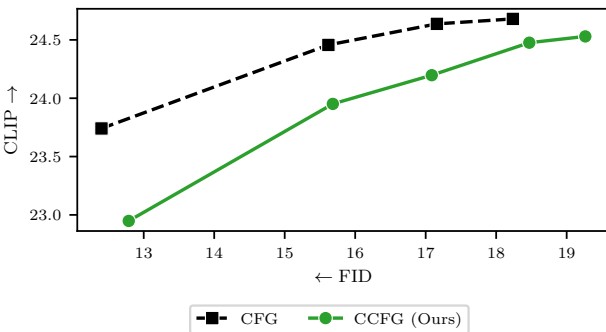

*Figure 26.* **Quantitative evaluation of CCFG and CFG.** CCFG achieves comparable FID scores to CFG while achieving slightly lower fidelity to the input prompts. However, unlike CFG, CCFG enables decompression without access to the original conditioning inputs.

bit-streams produced by CCFG does involve accessing the original input $\mathbf{y}$, and so our loss in CLIP scores are expected due to the rate-perception-distortion tradeoff (Blau & Michaeli, 2019) (here, we achieve $\frac{1000 \cdot \log_2(64)}{768^2} \approx 0.01$ BPP). Note that using CCFG in DDCM is fundamentally different than using CFG (Sec. 4), since the latter requires access to $\mathbf{y}$.

## C.7. Compressed Text-Based Image Editing

Image editing has seen significant progress in recent years, particularly in solutions that rely on pre-trained DDMs. Generally speaking, most methods rely on inversion (Huberman-Spiegelglas et al., 2024; Manor & Michaeli, 2024; Wu & De la Torre, 2023; Wallace et al., 2023; Hertz et al., 2023a), noising and denoising the original input (Meng et al., 2022), or manipulating the DDM's attention layers (Hertz et al., 2023b; Tumanyan et al., 2023). We demonstrate a simple approach for editing images using DDCM with a text-conditional DDM.

Specifically, we compress an image while feeding the DDM with a source prompt $c_{\text{src}}$ describing the image. To edit the image, we start by decoding it up to timestep $T_{\text{edit}}$ while feeding the DDM with the original prompt $c_{\text{src}}$. We then continue with the decoding from timestep $T_{\text{edit}}$, while feeding the DDM with the target prompt $c_{\text{dst}}$. Intuitively, this approach should preserve the low frequency contents in the original image (e.g., objects), while guiding the image towards the described edit.

We compare our approach to DDPM inversion (Huberman-Spiegelglas et al., 2024) and DDIM inversion (Song et al., 2021). For DDPM inversion we set $T = 100, T_{\text{skip}} = 36$, as recommended by the authors. For DDIM inversion we set $T = 100$, and follow Huberman-Spiegelglas et al. (2024) to additionally apply DDIM inversion mid-way, using $T_{\text{skip}} = 40$. For our approach, we set $T = 1000$ and $T_{\text{edit}} = 600$. Note that this value of $T_{\text{edit}}$ is equivalent to $T_{\text{skip}} = 40$ if $T = 100$ sampling steps were used. For all methods we use the pre-trained Stable Diffusion 1.4 checkpoint from Hugging Face. Images are taken from the modified-ImageNetR-TI2I dataset (Huberman-Spiegelglas et al., 2024; Tumanyan et al., 2023).

Preliminary qualitative results are shown Fig. 27. Our approach is less structure preserving than DDPM inversion, while offering more semantic object preservation than DDIM inversion. We encourage future works to investigate DDCM-based image editing methods.

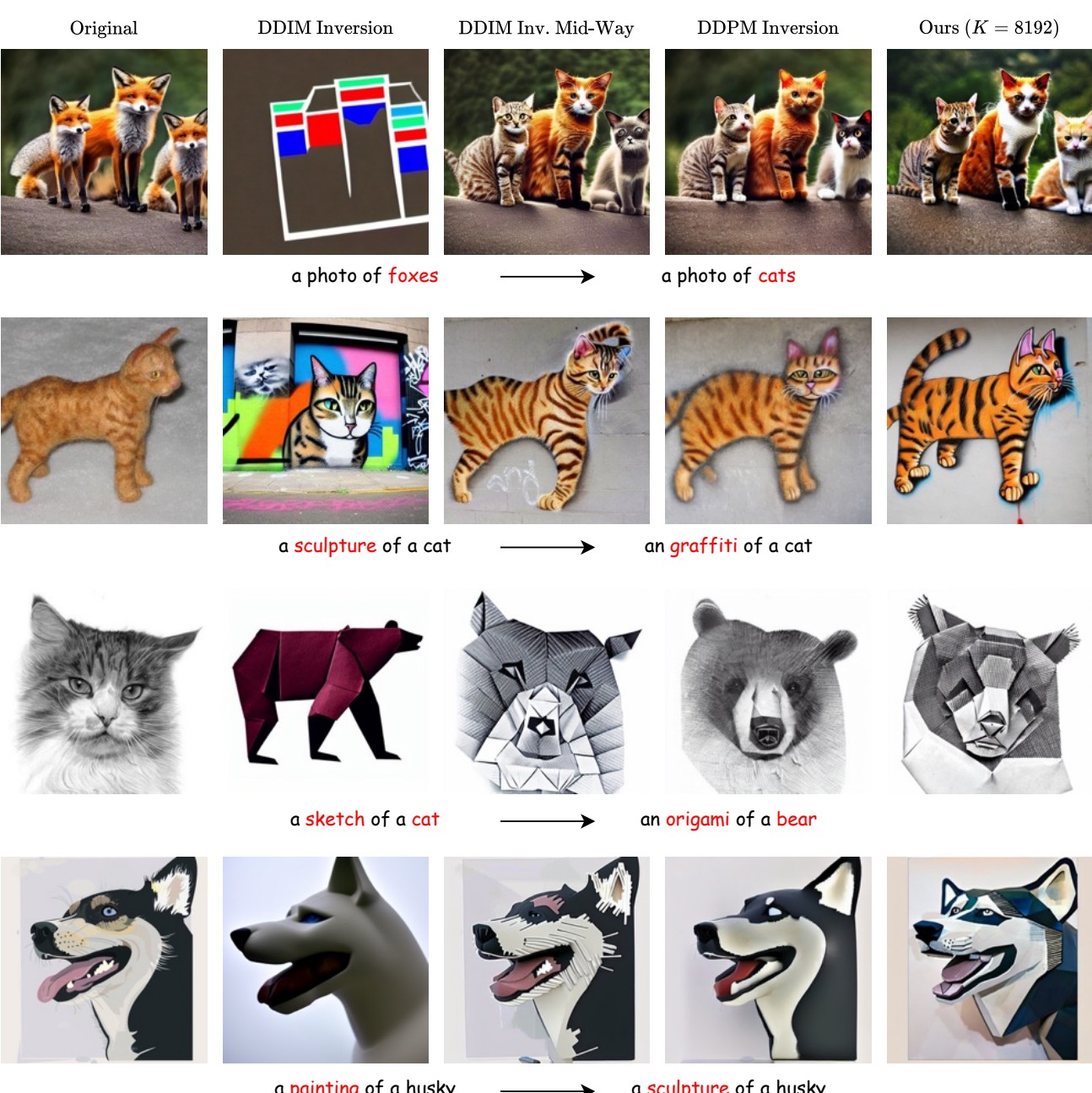

*Figure 27.* **Qualitative comparison of image editing methods.** Our approach preserves less image structure compared to DDPM inversion, while offering more semantic object preservation than DDIM inversion. This can be quite useful in scenarios where the editing prompt requires major structural changes, such as transforming a sketch of a cat into an origami of a bear.

