# OpenReview forum: "Compressed Image Generation with Denoising Diffusion Codebook Models"
_ICML.cc/2025/Conference — ICML 2025 poster_

### Official Review · Reviewer_r3bs · 2025-02-26

**Overall Recommendation:** 4

**Summary:**

This paper introduces a novel approach called the Denoising Diffusion Codebook Model, which replaces the standard Gaussian noise sampling in the reverse diffusion process with a codeword from a predefined codebook. This method enables the development of new lossy image codecs and, more broadly, compressed image restoration schemes.

**Claims And Evidence:**

Variants of the proposed Denoising Diffusion Codebook Model have a wide range of applications. This paper demonstrates them through image compression, compressed image restoration, and compressed text-based image editing, among others. Overall, the experimental results are quite convincing.

**Essential References Not Discussed:**

When it comes to compressed image restoration, the following paper should be cited and discussed.

H Liu, G Zhang, J Chen, AJ Khisti, Lossy compression with distribution shift as entropy constrained optimal transport, International Conference on Learning Representations, 2021

**Experimental Designs Or Analyses:**

The experimental designs are well aligned with the proposed theoretical framework.

**Methods And Evaluation Criteria:**

Yes, the proposed method, which replaces continuous Gaussian samples with discrete codewords, is intuitively clear.

**Other Comments Or Suggestions:**

The current version is acceptable. However, the paper would be stronger with a deeper analysis of a specific topic, such as image compression, rather than providing a uniform treatment of multiple applications.

**Other Strengths And Weaknesses:**

Overall, the proposed method is highly promising and has numerous potential applications. However, as noted at the end of the paper, several aspects of the method can be further improved. Additionally, there is a lack of theoretical understanding regarding its effectiveness.

**Questions For Authors:**

In image compression, is there any benefit to adjust the codebook size K over the timesteps?

**Relation To Broader Scientific Literature:**

This work is related to several areas, including generative models, image compression, and image restoration.

**Theoretical Claims:**

I have reviewed the theoretical analyses in the supplementary material. The analyses are relatively straightforward, and I have no concerns about their correctness.

---

> ### Author Rebuttal · Authors · 2025-03-29
>
> # Essential Reference
> We thank the reviewer for highlighting the insightful work by Huan Liu et al. (ICLR 2022). It is compelling to address tasks such as compressed image restoration by formalizing them as a distribution shift (via optimal transport) with an informational bottleneck. We will include a discussion of this important reference in the related work section of our revised manuscript.
>
> # Codebook Size Adjustment in Different Timesteps
> Dynamically adjusting the codebook size across timesteps may indeed offer potential benefits. A positive indication of this can be seen in Appendix B.3, where using different (fixed, not dynamically adjusted) codebook sizes for different timesteps leads to improved performance. However, identifying the “optimal” codebook size for each step can be challenging, as modifying the codebook size at some timestep can affect the outcomes of all subsequent timesteps. We will include this in the discussion as an interesting option for future research.

---

> > ### Comment · Reviewer_r3bs · 2025-04-02
> >
> > I have no further comments and will maintain the original rating.

---

### Official Review · Reviewer_bEwf · 2025-03-14

**Overall Recommendation:** 3

**Summary:**

This paper introduces Denoising Diffusion Codebook Models (DDCM), a approach that replaces standard Gaussian noise sampling in Denoising Diffusion Models (DDMs) with selections from fixed codebooks of i.i.d. Gaussian vectors. Despite using a discrete and finite noise representation, DDCM preserves the sample quality and diversity of standard DDMs. The method enables state-of-the-art perceptual image compression by selecting optimal noise samples for a given image and generalizes to other conditional generation tasks, such as image restoration. Additionally, the paper provides a mathematical interpretation linking DDCM to score-based posterior sampling.

**Claims And Evidence:**

Without rigorous inspection, it can be considered that the claims in this paper are all supported by evidence.

**Essential References Not Discussed:**

No essential references are not discussed.

**Experimental Designs Or Analyses:**

Just from the text expounded in this article, the experimental designs and analysis are somewhat reasonable.

**Methods And Evaluation Criteria:**

The paper defines bit rate based on the codebook size and the number of sampling timesteps, with the bit-stream length determined by their logarithmic relationship. This approach is closer to VQGAN-style methods rather than traditional compression metrics based on actual file size. While this provides a useful measure within the proposed framework, it would be helpful to clarify how this definition translates to real-world storage and transmission costs, and how it compares to standard rate-distortion metrics used in compression research.

**Other Comments Or Suggestions:**

See above.

**Other Strengths And Weaknesses:**

The idea of using discrete noise representations in diffusion models is intriguing, but the experimental setup and results are somewhat unclear. While the approach offers a novel take on generative modeling, its compression performance does not appear particularly strong when compared to traditional methods. As a generation method, it would benefit from more comprehensive evaluation using non-reference metrics, which could provide a clearer picture of its capabilities. The strength lies in the concept, but the execution in terms of compression and performance metrics needs further refinement.

**Questions For Authors:**

See above.

**Relation To Broader Scientific Literature:**

The paper is situated within the broader literature on diffusion models, generative modeling, and neural compression.

**Theoretical Claims:**

The theoretical part of this paper is fine.

---

> ### Author Rebuttal · Authors · 2025-03-29
>
> # DDCM Bitrate and File Size Clarification
> Please note that the reported bitrate is precisely that of the compressed file size, as in traditional compression methods. Indeed, the mentioned logarithmic relationship $T\cdot log_{2}(K)$ between the codebook size $K$ and the number of sampling steps $T$ directly calculates the file size (in bits). This holds since each DDCM-generated image corresponds to a sequence of $T$ integers, each in $[0,K-1]$, that represent the indices of the codebook entries chosen during the generation (as depicted in Fig. 2). This sequence is then represented in a binary format and saved as a file. E.g., for $T=3,K=4$, the indices sequence [1,3,0] translates to the bitstream “011100”. We compare our method against previous compression approaches (including traditional ones) according to their file size (using BPP).
> # Experimental Setup and Results Clarification
> We apologize for any confusion about our experimental setup and results. Our paper contains multiple different experimental settings (starting at lines 157(R), 188(R), 309(R), and 368(R)), involving image generation, compression, and restoration, following common practices in each of these fields. Let us clarify.
> ### Section 4 in the paper
> This section shows that DDCM maintains competitive generation capabilities relative to DDPM, even with small codebooks (lines 158(R)). Specifically, we show that DDCM achieves comparable performance to DDPM as measured by standard generative metrics like FID (Fig. 3, main text), KID (Fig. 8, App. A), and through visual inspection (Figs. 9 & 10, App. A) across two distinct datasets: ImageNet and MS-COCO. This experiment is intended to validate our hypothesis regarding the redundancy of the continuous Gaussian representation space utilized in traditional DDMs (lines 016(R) & 182(L)).
> ### Section 5 in the paper
> This section introduces our perceptual image compression method, based on DDCM. We compare our method to prior approaches using standard rate-perception-distortion evaluations [R2], assessing perceptual quality (FID) and distortion (LPIPS, PSNR) across multiple bitrates (BPP). These quantitative comparisons are provided in Fig. 5, complemented by qualitative assessments in Fig. 4, following common evaluation practices. Both our quantitative and qualitative results demonstrate the superiority of our proposed method over existing techniques (lines 243-258(L)). In Appendix B, we also include additional qualitative results, as well as additional quantitative results for higher-resolution images (Figs. 11-16).
> ### Section 6 in the paper
> Here, we propose extending our compression framework to handle more general compressed conditional generation tasks, such as compressed image restoration. Our experiments in the main text cover two tasks: zero-shot posterior sampling (Sec. 6.1) and blind face image restoration (Sec. 6.2). We compare our method with previous state-of-the-art approaches, assessing both distortion (PSNR) and perceptual quality (FID), thus adhering to established evaluation protocols in image restoration [R1]. The qualitative and quantitative results, presented in Figs. 6 & 7, show that our method achieves superior perceptual quality compared to previous methods (lines 353-357(L) & 421-428(L)). Apps. C.3 & C.4 provide additional results (Figs. 17-22).
>
> We hope this clearly addresses the concerns raised regarding the clarity and comprehensiveness of our experimental setup and results.
> # Superior Perceptual Compression Performance
> Kindly note that our image codec is designed for **perceptual** image compression, meaning that we aim to achieve best output perceptual quality. Due to the rate-perception-distortion tradeoff [R2], this is expected to compromise the distortion (e.g., PSNR, LPIPS) of our method. Indeed, since our method achieves the lowest (best) FID (perceptual quality) for almost all bitrates and datasets (Figure 5), the rate-perception-distortion tradeoff explains why our method does not always achieve the best distortion. However, note that we do achieve better distortion compared to previous perceptual image compression methods such as PerCo (SD), while also achieving better perceptual quality.
> # Generation Evaluation Using No-Reference Metrics
> We kindly note that it is not common practice to evaluate image generation methods with no-reference quality measures (e.g., NIQE [R3]), since the goal in image generation is to sample from a data distribution. No-reference quality measures only assess the quality of the generated images and not their diversity, whereas metrics such as FID and KID (which we report in the paper) assess quality and diversity simultaneously.
> ## References
> [R1] Yochai Blau & Tomer Michaeli. The perception-distortion tradeoff. CVPR 2018.
>
> [R2] Yochai Blau & Tomer Michaeli. Rethinking lossy compression: The rate-distortion-perception tradeoff. ICML 2019.
>
> [R3] Anish Mittal et al. Making a ‘completely blind’ image quality analyzer. ACCSC 2012.

---

> > ### Comment · Reviewer_bEwf · 2025-04-05
> >
> > The rebuttal addressed most of my concerns.

---

### Official Review · Reviewer_EqE9 · 2025-03-15

**Overall Recommendation:** 3

**Summary:**

This paper presents a novel approach, DDCM, to represent an image by DDPM procedure using a set of gaussian noise (codebook) indices. In other words, this paper shows that it is possible to "discretize" the "z" at every step of DDPM to approximate a high-quality diffusion procedure. DDCM can also be leveraged to solve various tasks, like text-to-image and image-to-image problems, which shed a light for codebook-based diffusion sampling procedure.

**Claims And Evidence:**

Claims in this submission are fully supported by clear and convincing evidence.

**Essential References Not Discussed:**

I have not found essential references not discussed in this submission. However, I feel it is necessary to include discrete visual tokenization methods in the discussion since the proposed DDCM essentially does the discretization on images using diffusion models. The authors can discuss DDCM and discrete tokenization methods in a paragraph but not experiments needed.

**Experimental Designs Or Analyses:**

The experimental designs and analyses are sound but some points and result presentation seem confusing.

1. In Figure 3 left, is the FID scale 10^1? Then it means that FID only decreases from 10.5 to 9.5 when increasing K from 2 to 64, which is not significant. Additionally, what is the exact FID value for DDPM in this figure? In my view, an FID of approximately 9.0 on ImageNet for DDPM or DDCM is not that strong. However, the visualizations do show high-quality samples. I'd like to see the authors' insights on this.

2. Relevant to P.1, what is the scale in Figure 5 FID, is it linear scale or log scale? Why show VAE bound instead of DDPM bound in Figure 5?

3. Can authors include the computing time of Equ (7)? Is the method efficient?

**Methods And Evaluation Criteria:**

The proposed method, DDCM, is suitable for the visual generative problem, especially for compressed image generation. The evaluation criteria, computing several metrics on validation set like ImageNet, makes sense for visual generation. There are some minor points and see experiment design parts.

**Other Comments Or Suggestions:**

Suggestions: the authors can provide absolute metric values in *numeric forms* in figure or table presentation, which could make results of DDCM more clear.

**Other Strengths And Weaknesses:**

Overall, this submission is a good and novel paper on discretizing the z noise in diffusion steps and representing images by compressed noise indices. For other minor concerns, please see Questions For Authors.

**Questions For Authors:**

I'd like to see insights for several open-end questions listed below from authors. No experiments needed.

1. Flow matching/rectified flow generative models are gaining popularity in the research community and have demonstrated competitive results. However, there is no random noise in ODE process behind FM except for initial point. Given this, can DDCM be applied to recent flow matching models like FLUX?

2. Is DDCM limited to the fixed-resolution generation process, given that the codebook fixed after intialization? Can noise codebook generalize to other resolutions with minimal efforts?

3. For better performance, is it doable to make noise map codebook learnable?

**Relation To Broader Scientific Literature:**

I think that the key contribution of this paper is discretization of the z noise in DDPM procedure, which is indeed novel and unexplored before.

**Theoretical Claims:**

I am not an expert on mathematical proof on diffusion theory but proofs in C.2 seem sound and correct to me.

---

> ### Author Rebuttal · Authors · 2025-03-29
>
> # Fig. 3 FID
> Yes, the FID scale in Fig. 3-L is $10^1$. The FID of DDPM is 9.21 (black dashed line). Indeed, some prior works with the same DDM reported lower FID on ImageNet 256x256, when using 50k generated samples [R2]. However, we use only **10k generated samples** (see L171-L) to reduce computation time, similarly to [R3,R4]. Decreasing the number of samples is known to increase the FID [R1]. To validate this, we computed the FID between the same 50k reference images and two random subsets from the ImageNet 256x256 training set, consisting of 10k and 50k images. These subsets obtained FID=4.934 and FID=1.968, respectively.
>
> Note that this experiment only aims to demonstrate DDCM's competitiveness with DDPM, specifically for small codebooks. Indeed, the modest decrease in FID from 10.5 to 9.5 (when increasing $K$) is a strength of our method rather than a limitation: even for $K=2$, DDCM achieves FID$\approx$10.5 (competitive with DDPM). This supports our hypothesis (L016-R & L182-L) that the Gaussian representation space used by standard DDMs is redundant.
> # Fig. 5 & VAE Bound
> The FID scores for the compression results in Fig. 5 are in log scale.
>
> Regarding the “VAE bound,” we apologize for the misunderstanding, this term was not clarified in the paper. In some experiments we employ a latent DDM (specifically, SD 2.1), so we are compressing the latent VAE encoding of a given image, rather than the image itself. Since encoding-decoding an image using a VAE typically does not yield perfect image reconstruction, the distortion of any VAE-latent-space-based compression method (e.g., ours, PerCo, PSC) is bounded by that of the VAE encoder-decoder. The “VAE bound” in the figure corresponds to the distortion resulting from encoding and decoding the original images using only the VAE, **without any additional compression**. This bound is important to present since it allows to distinguish the distortion caused by the VAE from that caused by the compression scheme.
>
> Thanks for these important points. We will clarify them in the revised paper and replace the term “VAE-bound” with “SD 2.1 Encoder-Decoder bound.”
> # Eq. (7) Efficiency
> Please note that Eq. (7) does not involve any gradient computation, iterative optimization, or use of a neural network. It involves a straightforward matrix multiplication followed by picking the maximum out of K scalars. Empirically, on an L40S GPU with $K=8192$ and 256x256 images, this operation takes 0.357ms on average (over 100 trials). This is negligible compared to the DDM’s forward pass, which takes 57.2ms on average.
> # Tokenization Methods
> We appreciate this suggestion. Viewing DDCM-based compression as an image tokenization method with random noises acting as the tokens is interesting. We will discuss this in our paper revision.
> # Numerical Results
> Thanks for this valuable suggestion. We will add numerical comparisons to the appendix.
> # Applying to Flow Matching Models
> Although flow-matching models typically generate samples by solving an ODE, several works (e.g., [R5,R6]) have shown they can also generate samples by solving an SDE, which involves adding noise at each generation step. Thus, we believe that such flow-matching SDEs may be discretized as well, similarly to DDCM.
> # Resolution Generalization
> DDCM is easily generalizable to any resolution. Instead of pre-sampling the codebooks with a particular resolution, one can rely on a shared-seed random generator to dynamically generate codebooks for different resolutions. Specifically, one can store/transmit the resolution as part of the bitstream and, before compressing/decompressing an image, randomly sample the codebooks for the given resolution. Using the same seed for compression and decompression ensures that both would rely on the same codebooks.
>
> If no shared-seed random generator is available, one can pre-sample the codebooks for a maximal allowed resolution, store/transmit the resolution of the image, and dynamically slice smaller codebooks from the pre-sampled ones.
>
> Therefore, the generalizability of DDCM to different resolutions is determined only by that of the pre-trained DDM.
> # Learnable Codebooks
> Learning the codebooks may indeed enhance DDCM’s performance (as noted in L433-R), particularly in compression. We investigate this in our future work.
>
> ## References
> [R1] Mikołaj Bińkowski et al. Demystifying MMD GANs. ICLR 2018.
>
> [R2] Cheng Lu et al. DPM-solver: A fast ode solver for diffusion probabilistic model sampling in around 10 steps. NeurIPS 2022.
>
> [R3] Cheng Lu et al. DPM-Solver++: Fast solver for guided sampling of diffusion probabilistic models. arXiv:2211.01095.
>
> [R4] Kaiwen Zheng et al. DPM-Solver-v3: Improved diffusion ODE solver with empirical model statistics. NeurIPS 2023.
>
> [R5] Jeongsol Kim et al. FlowDPS: Flow-driven posterior sampling for inverse problems. arXiv:2503.08136.
>
> [R6] Litu Rout et al. Semantic image inversion and editing using rectified stochastic differential equations. ICLR 2025.

---

### Decision · Program_Chairs · 2025-05-01

**Decision:**

Accept (poster)

**Comment:**

This paper has received all acceptances in the final recommendations. The authors present a novel approach, DDCM, to represent an image by the DDPM procedure using a set of Gaussian noise indices. Although the reviewers expressed concerns about the unclear experimental setup and suboptimal compression performance, the authors' rebuttal effectively addressed several of these issues. This led to a consensus among the reviewers in favor of acceptance. Consequently, the Area Chair has decided to accept this paper.